# Consistent Low-Rank Adaptation of Two-layer Neural Networks: A Nonparametric Statistics Approach

## Abstract

Low-Rank Adaptation (LoRA) is a powerful technique for fine-tuning Large Language Models (LLMs), offering greater parameter efficiency and improved generalization in data-constrained settings. While its advantages makes it highly promising for general transfer learning, its reliance on iterative optimization methods such as SGD still demands substantial computation and poses a challenge for theoretical analysis. We propose a novel two-step, closed-form approach for LoRA in two-layer feedforward neural networks (FNN) that mitigates the reliance on iterative algorithms. First, by leveraging Stein's lemma, a classical statistical tool, we derive an analytical estimator for the first-layer LoRA parameters. Second, we solve for the second-layer parameters via reduced-rank ridge regression. We provide theoretical guarantees for the low-rank parameter estimation under a projection adaptation assumption: the optimal first layer adaptation removes irrelevant directions via subspace projection. This generalizes the concept of rank pruning, which removes irrelevant low-rank components from a weight matrix. Crucially, our solution is non-iterative and computationally efficient, computing the full adaptation in seconds—a fraction of the time required by SGD-based LoRA. Numerical experiments on MNIST suggest that our method not only significantly reduces computational cost and achieves prediction performance comparable to that of a fully trained LoRA model, but also serves as a good initialization for SGD-based LoRA.

## 1 Introduction

Low-Rank Adaptation (Hu et al., 2022, LoRA) has become a powerful technique for efficiently fine-tuning Large Language Models (LLMs). By constraining weight updates to a low-rank subspace, LoRA reduces the number of trainable parameters, thereby lowering computational demands and enhancing generalization in data-scarce settings (Lin et al., 2024). While LoRA's empirical success in fine-tuning LLMs is evident, its core principle—exploiting low-dimensional structure in parameter updates—holds broader promise for transfer learning beyond language models. This paper explores application of this low-rank update principle to the adaptation of a pre-trained two-layer neural network.

Transfer learning is a core paradigm in machine learning that leverages knowledge from a data-rich source domain to improve performance in a data-scarce target domain (Zhuang et al., 2020). Its applications are broad, ranging from deploying ImageNet-pretrained models for specialized medical imaging tasks (Morid et al., 2021) to adapting BERT for targeted sentiment classification (Prottasha et al., 2022). Its importance is growing in modern AI settings that require not only effectiveness but also flexibility, including personalized modeling that tailors a base model to individual users with limited data (Yoon et al., 2017) and federated learning, where models are adapted on decentralized edge devices (Guo et al., 2024). Across these scenarios, training the full model is infeasible due to limited data. LoRA's parameter-efficient philosophy is ideally suited to these constraints. Moreover, LoRA's philosophy aligns with classical methods in high-dimensional statistics that exploit low-dimensional structure, such as robust PCA (Candès et al., 2011) and reduced-rank regression (Yuan et al., 2007), which are known to improve both statistical efficiency and interpretability.

Despite its practical appeal, standard LoRA implementations depend heavily on iterative optimizers such as stochastic gradient descent (SGD) and ADAM (Kingma & Ba, 2014), which introduce several limitations for transfer learning. First, the iterative process remains computationally demanding, often requiring multiple epochs over the data. Second, it obscures the statistical properties of the resulting LoRA estimator, making it difficult to establish theoretical guarantees such as consistency. Third, the training time constrains rapid adaptation in scenarios that require instant updates, such as real-time personalization or on-device learning.

To overcome these limitations, we introduce a novel, non-iterative framework for LoRA-style adaptation of two-layer feedforward networks. Our method computes a closed-form, two-step estimator that mitigates the reliance on iterative training algorithms such as SGD. Our method first leverages Stein's lemma, a classical tool from statistical estimation, to derive an analytical estimator for the first-layer low-rank parameters. In the second step, we solve for the second-layer parameters via a reduced-rank ridge regression (Mukherjee & Zhu, 2011), which has a elegant and interpretable closed-form solution. This two-step procedure is deterministic, requires no learning rate tuning or convergence checks, and computes the full model adaptation in a single pass.

We establish theoretical consistency of our estimator under a projection adaptation assumption, which posits that the optimal first-layer update acts as an orthogonal projection—removing irrelevant directions from the pre-trained weight. This generalizes rank pruning and provides a tractable framework for analysis. We further derive non-asymptotic convergence rates for the estimator and its excess risk, and characterize the bias-variance trade-off induced by regularization.

The main benefit of our approach is its dramatic computational efficiency. Our closed-form solution computes the full model adaptation in a matter of seconds—a fraction of the time required by SGD-based LoRA. In addition, it serves as a better initialization for regular SGD than the traditional zero initialized LoRA. We validate our method through numerical experiments on MNIST dataset, demonstrating that it achieves prediction performance comparable to SGD. By providing a fast, transparent, and theoretically sound alternative to SGD-based adaptation, this work paves the way for more efficient transfer learning algorithms. Our main contributions are summarized as follows:

1. **A closed-form, non-iterative solution to LoRA**: we introduce a novel approach that yields an explicit, closed-form solution to LoRA in two-layer neural networks, mitigating reliance on iterative algorithms such as SGD.

2. **Theoretical consistency guarantees**: we establish consistency of our closed-form estimators under a projection adaptation assumption, and provide non-asymptotic error bounds characterizing their convergence rate.

3. **Significant computational gains**: adaptation completes in seconds, enabling real-time or edge-device applications.

4. **Improved initialization for Iterative Algorithms**: our estimation outperforms zero initialization, accelerating fine-tuning.

5. **Empirical validation on MNIST**: our approach attains predictive performance comparable to iterative LoRA, while delivering full transparency and significant speedups.

## 2 PROBLEM SETUP

We consider a two-layer neural network model:

$$Y_i = (W_2 + \Delta_2)\, a\big((W_1 + \Delta_1)\phi(X_i)\big) + \epsilon_i. \tag{1}$$

Here, $X_i \in \mathbb{R}^p$ is a high-dimensional input and $Y_i \in \mathbb{R}^q$ is its corresponding output. A key component of our setup is a fixed feature map $\phi : \mathbb{R}^p \to \mathbb{R}^{l_1}$, which projects the input into a lower-dimensional representation space (e.g., from a pre-trained encoder). The activation function $a : \mathbb{R} \to \mathbb{R}$ acts component-wise, and $\epsilon_i$ is an independent zero-mean noise term. The matrices have dimensions $W_1, \Delta_1 \in \mathbb{R}^{l_2 \times l_1}$ and $W_2, \Delta_2 \in \mathbb{R}^{q \times l_2}$. We call $(W_1, W_2)$ the prior weight matrices, and $(\Delta_1, \Delta_2)$ the adaptation matrices. The prior weight matrices, which serve as our initialization, encode knowledge from a source task or a large foundational dataset. Our goal is to adapt this model to a new target task by learning a pair of low-rank matrices $(\Delta_1, \Delta_2)$, while keeping $(W_1, W_2, \phi)$ fixed. This type of update, famously effective in fine-tuning large language models, is known as Low-Rank Adaptation (LoRA).

The rank constraint on $(\Delta_1, \Delta_2)$ offers several key advantages. First, the number of trainable parameters is reduced, which enhances computational efficiency and mitigates overfitting. Second, the low-rank update captures the most significant task-specific directions in the parameter space, promoting generalization in data-constrained settings and mitigating catastrophic forgetting by remaining close to the prior model.

Analysis of Model (1) has significant practical implications. The assumption that high-dimensional data lies on a low-dimensional manifold is foundational to deep learning. The representation map $\phi : \mathbb{R}^p \to \mathbb{R}^{l_1}$ is typically a pre-trained encoder (e.g., from an autoencoder, VAE (Kingma, 2013), or WAE (Tolstikhin et al., 2017)) that projects raw inputs into a semantically meaningful latent space. This allows even a shallow network to model complex functions by operating on these distilled representations. In addition, the latent representations $\phi(X_i)$ can be regularized to follow a certain distribution, such as standard gaussian, which can help in downstream tasks. The consideration of representations as input has practical implications in our model, as will be discussed later.

To handle high-dimensional outputs $Y_i \in \mathbb{R}^q$, we may similarly introduce an output representation $\psi : \mathbb{R}^q \to \mathbb{R}^d$, leading to the generalized model: $\psi(Y_i) = (W_2 + \Delta_2)\, a\big( (W_1 + \Delta_1)\phi(X_i) \big) + \epsilon_i$. This framework of mapping between two latent spaces is a powerful and common paradigm in deep learning (Maiorca et al., 2023; Lähner & Moeller, 2024; Insulla et al., 2025). It has found applications in multi-modal learning (Insulla et al., 2025), domain translation (Lähner & Moeller, 2024) and image generation (Maiorca et al., 2023).

Unlike standard empirical risk minimization, our objective is the exact recovery of the true adaptation matrices $(\Delta_1, \Delta_2)$. We show this is theoretically possible under a Projection Adaptation Assumption, which posits that the optimal adaptation in the first layer primarily involves projecting input representations onto a task-relevant subspace. In other words, it suggests there exists an orthogonal projection matrix $P$ such that $\mathbb{E}[Y|P\phi(X)] \approx f(X)$. The assumption can formally be defined as follows:

**Assumption 2.1.** *(Projection Adaptation) Let $f_\theta : \mathbb{R}^{l_1} \to \mathbb{R}^q$ be a prior model with parameter $\theta$, and let $\Gamma$ be a constrained parameter space. Assume there exists an orthogonal projection matrix $P = VV^T \in \mathbb{R}^{l_1 \times l_1}$, where $V \in \mathbb{R}^{l_1 \times k_0}$ has orthonormal columns and $k_0 < l_1$, and a parameter $\gamma \in \Gamma$ such that*

$$\mathbb{E}[Y|\phi(X)] = f_{\theta+\gamma}\left(P\phi(X)\right). \tag{2}$$

*An example of $\Gamma$ is the space of low-rank matrices.*

This assumption is frequently justified in practice. For instance, the Linear Representation Hypothesis (Park et al., 2023; Mikolov et al., 2013; Wang et al., 2023) suggests that human-interpretable concepts are encoded in linear subspaces of representation spaces. Fine-tuning can thus be viewed as suppressing irrelevant concepts (by projecting them out) and amplifying task-relevant ones (Uppaal et al., 2024). This perspective provides a geometric interpretation for manipulating model behavior through linear projections. A special case of this assumption is rank-pruning of a weight matrix. When $P$ is defined by a subset of the right singular vectors of $W_1$, the update $W_1 P$ effectively removes some rank-one components of the weight matrix.

Building upon this observation, our work introduces a consistent method for identifying the optimal orthogonal projection and updating the low-rank matrices. In the subsequent sections, we will detail our estimation procedure, provide theoretical guarantees for parameter recovery.

**Notation:** Suppose $A \in \mathbb{R}^{n \times m}$ is a rank $r_A$ matrix with Singular Value Decomposition (SVD) of the form $A = USV^T$, where $U = \left[U_k, U_k^\perp\right] \in \mathbb{R}^{n \times r_A}, U_k \in \mathbb{R}^{n \times k}$, $S \in \mathbb{R}^{r_A \times r_A}$ is diagonal, $V = \left[V_k, V_k^\perp\right] \in \mathbb{R}^{m \times r_A}$, and $V_k \in \mathbb{R}^{m \times k}$. We write $\mathrm{SVD}_r(A) = V$ and $\mathrm{SVD}_l(A) = U$. Moreover, we use $\mathcal{P}_k^r(A) = V_k V_k^T$ to denote the orthogonal projection matrix onto the subspace spanned by its top-$k$ right singular vectors, $\mathcal{P}_k^l(A) = U_k U_k^T$ for the top-$k$ left singular vectors, and $\mathcal{R}_k(A) = [U_k, 0]S[V_k, 0]^T$ to denote its rank-$k$ approximation. We use $\|A\|_F$ to denote its Frobenius norm, $\|A\|_{op}$ to denote its operator norm, and $\mathrm{rank}(A)$ to denote its rank. For a vector $v \in \mathbb{R}^n$, we use $\|v\|_2$ to denote its $\ell_2$-norm. For two sequences, we say that $a_n \lesssim b_n$ (also written as $a = \mathcal{O}(b_n)$) if there exists $C > 0$ such that $a_n \leq Cb_n$.

## 3 Model Estimation with Stein's Lemma and Subspace Projection

Assume $\text{rank}(\Delta_i) = r_i$ for $i = 1, 2$. Denote the $j$-th element of the target vector $Y_i$ as $Y_{ij}$ for notational simplicity. Denote the latent representation as $Z_i = \phi(X_i)$, and its transformed version as $\widetilde{Z}_i = (W_1 + \Delta_1)Z_i$. We assume $Z_i$ is distributed according to a known density $p(z)$, which is often enforced in practice. For instance, in WAEs, $Z_i$ is usually enforced to be $\phi(X_i) \sim \mathcal{N}(0, I)$. This allows us to define the known second-order score function:

$$S(z) = T(z)T(z)^T - \nabla T(z), \tag{3}$$

where $T(z) = \nabla p(z)/p(z)$ is the first-order score function. For a standard Gaussian $p(z) = \mathcal{N}(0, \Sigma)$, this simplifies to $S(z) = \Sigma^{-1}zz^T\Sigma^{-1} - \Sigma^{-1}$. In our experiments, we show that using a Gaussian approximation for the score function when the true score is unknown still delivers satisfactory performance. Finally, we define the function $f : \mathbb{R}^{l_2} \to \mathbb{R}^q$ for the adapted second layer as $f(\widetilde{z}) = (f_1(\widetilde{z}), \ldots, f_q(\widetilde{z}))^T = (W_2 + \Delta_2)a(\widetilde{z})$.

Our estimator is grounded in the following identity:

**Lemma 3.1.** *(Second-Order Stein's Identity) Suppose model (1) holds. For any $j \in [q]$, if the expectations $\mathbb{E}[Y_{ij}S(Z_i)]$ and $\mathbb{E}\left[\nabla^2_{\widetilde{Z}_i} f_j(\widetilde{Z}_i)\right]$ exists and are well-defined, and $\lim_{\|z\| \to \infty} f_j(\widetilde{z})p(z) \to 0$ and $\lim_{\|z\| \to \infty} \nabla_{\widetilde{z}} f_j(\widetilde{z})p(z) \to 0$, then*

$$\mathbb{E}[Y_{ij}S(Z_i)] = (W_1 + \Delta_1)^T \mathbb{E}\left[\nabla^2_{\widetilde{Z}_i} f_j(\widetilde{Z}_i)\right] (W_1 + \Delta_1). \tag{4}$$

Lemma 3.1 implies that the row space of $\mathbb{E}[Y_{ij}S(Z_i)]$ is contained within the row space of $(W_1+\Delta_1)$ for each $j$. Therefore, a valid estimator for this space can be obtained via the Eigenvalue decomposition of the empirical mean: $\widehat{V}_j = \text{SVD}_l\left(\frac{1}{n}\sum_{i=1}^n Y_{ij}S(Z_i)\right)$. Since each $\widehat{V}_j$ estimates a basis for the same row space, we combine them for statistical efficiency. We adopt the straightforward Stack-SVD method (Baharav et al., 2025):

$$\widehat{V} = \text{SVD}_l\left(\frac{1}{n}\left[\sum_{i=1}^n Y_{i1}S(Z_i), \ldots, \sum_{i=1}^n Y_{iq}S(Z_i)\right]\right). \tag{5}$$

Let $V$ be a matrix whose columns are the right singular vectors of $(W_1 + \Delta_1)$. From the identity $W_1 + \Delta_1 = (W_1 + \Delta_1)VV^T$, we use below equality to build our estimator of $\Delta_1$:

$$W_1\left(I - VV^T\right) + \Delta_1\left(I - VV^T\right) = 0. \tag{6}$$

With equation (6) and $\widehat{V}$ in (5), we propose the following problem to estimate $\Delta_1$,

$$\min_{\Delta_1 \in \mathbb{R}^{l_2 \times l_1}} \left\|W_1(I - \widehat{V}\widehat{V}^T) + \Delta_1(I - \widehat{V}\widehat{V}^T)\right\|_F^2 + \lambda_1\|\Delta_1\|_F^2, \tag{7}$$

$$\text{subject to} \quad \text{rank}(\Delta_1) \leq k_1. \tag{8}$$

where $\lambda_1 > 0$ is a regularization parameter, and $k_1 \in \mathbb{Z}^+$ is the rank constraint. A key property of problem (7) is that it admits an explicit, closed-form solution, as shown in the proposition 3.2.

**Proposition 3.2.** *The global minimizer of the constrained minimization problem (7) is*

$$\widehat{\Delta}_1 = \frac{1}{1 + \lambda_1}\mathcal{R}_{k_1}\left(-W_1\left(I - \widehat{V}\widehat{V}^T\right)\right). \tag{9}$$

An intuitive interpretation follows: without rank constraints or regularization (i.e., $\lambda_1 = 0, k_1 = \min\{l_1, l_2\}$), the adapted mapping satisfies $\left(W_1 + \widehat{\Delta}_1\right)Z_i = W_1\widehat{P}Z_i$, where $\widehat{P} = \widehat{V}\widehat{V}^T$. In effect, this projects the input representations onto the estimated task-specific subspace, filtering out components irrelevant to the new task. Revisiting (6), the term $W_1VV^T$ projects the original weights onto the task-specific subspace, removing features irrelevant to the new task, while $\Delta_1VV^T$ encodes the task-specific information that must be learned. Because (6) offers no direct constraint on $\Delta_1VV^T$, the minimum-norm solution drives this term to zero. Notably, when the columns of $V$ are a subset of $W_1$'s right singular vectors, this reduces to rank pruning.

The regularization parameters control the adaptation strength and structure. The norm regularization $\lambda_1$ shrinks the adaptation towards zero, enforcing a soft bias to the original pre-trained weights $W_1$. In the limit $\lambda_1 \to \infty$, we have $\widehat{\Delta}_1 \to 0$, and the model effectively reverts to the frozen pre-trained prior. The rank constraint $k_1$ enforces a low-rank structure on $\Delta_1$, providing an inductive bias. When the true adaptation is low-rank (small $r_1$), this constraint improves statistical efficiency. We analyze this trade-off in Section 4.

With learned $\widehat{\Delta}_1$, we now consider the problem of estimating the second-layer adaptation $\Delta_2$. We frame this as a regularized regression problem. Define the adapted activation matrix: for each data point, let $\widehat{A}_i = a\left((W_1 + \widehat{\Delta}_1)Z_i\right)$ and stack them into the matrix $\widehat{A} = (\widehat{A}_1, \ldots, \widehat{A}_n)^T$. Let $Y = (Y_1, \ldots, Y_n)^T$ be the target matrix. The goal is to find a low-rank $\Delta_2$ such that $\widehat{A}(W_2 + \Delta_2)^T$ approximates $Y$. This leads to the following optimization problem.

$$\min_{\Delta_2 \in \mathbb{R}^{q \times l_2}} \quad \left\|Y - \widehat{A}(W_2 + \Delta_2)^T\right\|_F^2 + \lambda_2 \|\Delta_2\|_F^2,$$
$$\text{subject to} \quad \text{rank}(\Delta_2) \leq k_2, \tag{10}$$

The problem (10) also has an explicit solution, as demonstrated in the following proposition.

**Proposition 3.3.** *The global minimizer of the constrained minimization problem (10) is*

$$\widehat{\Delta}_2 = \widehat{B}\widehat{P}_{k_2}, \tag{11}$$

*where* $\widehat{B} = \frac{1}{n}\left(\widehat{A}^T\widehat{A}/n + \lambda_2 I\right)^{-1}\widehat{A}^T\left(Y - \widehat{A}W_2^T\right)$ *and* $\widehat{P}_{k_2} = \mathcal{P}_{k_2}^r\left(\left(Y - \widehat{A}W_2\right)^T \widehat{A}\widehat{B}^T\right)$. *Here,* $\mathcal{P}_{k_2}^r(\cdot)$ *denotes projection matrix onto subspace spanned by the top-$k_2$ right singular vectors of its argument.*

The estimator $\widehat{\Delta}_2$ has an intuitive structure. The matrix $\widehat{B}$ is the standard ridge regression coefficient matrix for predicting the residual targets $(Y - \widehat{A}W_2^T)$ from the adapted features $\widehat{A}$. The projection matrix $\widehat{P}_{k_2}$ then projects this solution onto a subspace, ensuring $\text{rank}(\widehat{\Delta}_2) \leq k_2$. With learned $\widehat{\Delta}_2$, the predictions are $\widehat{A}(W_2 + \widehat{\Delta}_2)^T \approx \widehat{A}W_2^T + \mathcal{R}_{k_2}(\widehat{A}B^T)$, where $\mathcal{R}_{k_2}(\cdot)$ denotes the best rank-$k_2$ approximation of its argument. Thus, the adaptation adds the best rank-$k_2$ approximation of the ridge predictions for the residuals to the original model's predictions.

We summarize our two-step LoRA estimation procedure as below:

---

**Algorithm 1** LoRA tuning with Stein's Lemma and subspace projection

---

**Require:** $(W_1, W_2, \phi)$, $S(z)$, $(\lambda_1, \lambda_2, k_1, k_2)$, $\{X_i, Y_i\}_{i=1}^n$
 1: Subspace estimation via Stein's Lemma to obtain $\widehat{V}$ as in (5).
 2: Compute closed-form solution of $\widehat{\Delta}_1$ using $\widehat{V}$ as in (9).
 3: Compute closed-form solution of $\widehat{\Delta}_2$ as in (11).
 4: **return** $\widehat{f}(x) = \left(W_2 + \widehat{\Delta}_2\right)a\left(\left(W_1 + \widehat{\Delta}_1\right)x\right)$.

---

We conclude this section with a few remarks. First, under the projection adaptation assumption, when the target function takes the form $f_{W_2+\gamma}(x) = (W_2 + \gamma)a(W_1 P \phi(x))$ with a rank constraint on $\gamma$, our method could recover the target parameters consistently. This includes the identified orthogonal projection $\widehat{P} = \widehat{V}\widehat{V}^T$ and the output adaptation $\gamma = \widehat{\Delta}_2$. The assumption, which is equivalent to $\|\Delta_2 VV^T\|_F = 0$, asserts that the target task only requires projecting the first-layer representations $\phi(x)$. This is reasonable when the source model is well trained on a large, general-purpose dataset, since $W_1$ likely already captures most of the necessary feature transformations, making $\|\Delta_2 VV^T\|_F$ small.

Second, our dual regularization is well suited for transfer learning. A central challenge in transfer learning is mitigating negative transfer, where irrelevant source knowledge degrades target performance. Our hyper-parameters $\lambda_1$ and $\lambda_2$ provide explicit control over the degree of transfer by governing the shrinkage strength toward the original pre-trained weights. This allows practitioners to precisely calibrate how much source information to retain.

# 4 THEORETICAL ANALYSIS

In this section, we establish theoretical guarantees for the proposed two-step estimator. Our goal is to establish the consistency of our method and analyze the bias-variance trade-off induced by the tuning parameters $(\lambda_1, \lambda_2, k_1, k_2)$.

We begin by defining certain key quantities. Let $\delta = \|\Delta_1 VV^T\|_F$ measure the deviation from the projection adaptation assumption. Recall that $Z_i = \phi(X_i)$ denotes the latent representation, $\widetilde{Z}_i = (W_1 + \Delta_1)Z_i$ its transformed version, and $f(\widetilde{z}) = (W_2 + \Delta_2)a(\widetilde{z})$ the second layer function. Let $S(z)$ denote the second-order score function of $Z_i$. Additionally, let $r_i = \text{rank}(\Delta_i)$, $i = 1, 2$, denote the true rank of the adaptation matrix. Our analysis relies on the following assumptions:

**Assumption 4.1** (Non-singular Hessian). *There exists an output dimension $1 \leq j \leq q$ such that the expected Hessian $\mathbb{E}[\nabla^2_{\widetilde{z}} f_j(\widetilde{Z})]$ is non-singular.*

**Assumption 4.2** (Boundedness). *There exist constants $c_1, c_2, c_3 > 0$ such that $\|Z_i\|_2 \leq c_1, \|\epsilon_i\|_2 \leq c_2$ and $\|S(Z_i)\|_{op} \leq c_3$ almost surely.*

**Assumption 4.3** (Lipschitz activations). *The activation function $a : \mathbb{R} \to \mathbb{R}$ is a $K$-Lipschitz continuous.*

Assumption 4.1 is mild. To see this, let $w_{j,\cdot}$ be the $j$-th row of $W_2 + \Delta_2$. The expected Hessian is diagonal: $\mathbb{E}[\nabla^2_{\widetilde{z}} f_j(\widetilde{z})] = \text{diag}\,(w_{j,1}\mathbb{E}[a''(\widetilde{z}_1)], \ldots, w_{j,l_2}\mathbb{E}[a''(\widetilde{z}_{l_2})])$. For ReLU, $\mathbb{E}[a''(\widetilde{z}_k)]$ corresponds to the density at zero, so the assumption holds as long as at least one row of $W_2 + \Delta_2$ does not contain zeros . Assumption 4.2 is standard in statistical learning and reasonable in practice for bounded features and noise. Assumption 4.3 holds for common activations including ReLU, Softplus, and more.

We evaluate the performance of the estimation via the excess risk:

$$\mathcal{E}(\widehat{\Delta}_1, \widehat{\Delta}_2) := L(\widehat{\Delta}_1, \widehat{\Delta}_2) - \inf_{\Delta_1, \Delta_2} L(\Delta_1, \Delta_2), \tag{12}$$

where the population risk is $L(\Delta_1, \Delta_2) := \mathbb{E}\left[\|y - (W_2 + \Delta_2)a((W_1 + \Delta_1)\phi(x))\|_2^2\right]$. Our main result characterizes the convergence rate of the excess risk:

**Theorem 4.1.** *(Excess risk) Assume the data-generating model (1) and Assumptions 4.1 - 4.3 hold. Then, for any unseen test point $(x, y)$, the excess risk satisfies*

$$\mathcal{E}(\widehat{\Delta}_1, \widehat{\Delta}_2) \lesssim \Delta_{stat} + \Delta_{first} + \Delta_{bias}, \tag{13}$$

*where*

$$\Delta_{stat} = k_2 \left(\frac{K_{\lambda_2}}{\lambda_2}\right)^2 \left(\frac{1}{n} + \sqrt{\frac{1}{n}\mathbb{E}\|\Delta_1 - \widehat{\Delta}_1\|_{op}^2}\right) + \left(1 + \frac{1}{\lambda_2 n}\right)\frac{Tr((\Sigma + \lambda_2)^{-1}\Sigma)}{n}Tr(\mathbb{E}[\epsilon\epsilon^T]P_{k_2}),$$

$$\Delta_{first} = \left(\left(1 + \frac{1}{\lambda_2}\right)^2 + C_{\lambda_2}^2 + k_2\left(\frac{K_{\lambda_2}}{\lambda_2}\right)^2\right)\mathbb{E}\|\Delta_1 - \widehat{\Delta}_1\|_{op}^2,$$

$$\Delta_{bias} = \|(I - P_{k_2})\Delta_2\Sigma^{\frac{1}{2}}\|_F^2 + \lambda_2\left(1 + \frac{1}{\lambda_2 n}\right)^2\|(\Sigma + \lambda_2)^{-\frac{1}{2}}\Sigma^{\frac{1}{2}}P_{k_2}\Delta_2\|_F^2.$$

*Here, $C_{\lambda_2} = \left(\frac{1}{\lambda_2} + \frac{1}{\lambda_2^2} + \frac{1}{\lambda_2^3}\right)$ and $K_{\lambda_2} = \left(1 + \frac{1}{\lambda_2}\right)$ are constants depending on $\lambda_2$, $\Sigma = Cov\left[a\left((W_1 + \Delta_1)\phi(X_i)\right)\right]$, and $P_{k_2} = \mathcal{P}_{k_2}^r\left(\Delta_2\Sigma(\Sigma + \lambda_2 I)^{-1}\Sigma\Delta_2^T\right)$ is the rank-$k_2$ projection matrix onto the row space of the indicated matrix..*

Theorem 4.1 reveals a bias-variance trade-off governed by the second-layer hyper-parameters $\lambda_2$ and $k_2$. First, $\Delta_{\text{first}}$ is the error due to the first-layer estimation. It vanishes if $\widehat{\Delta}_1$ is a consistent estimation for $\Delta_1$. Second, $\Delta_{\text{stat}}$ is the statistical error. If $\mathbb{E}\|\widehat{\Delta}_1 - \Delta_1\|_{\text{op}}^2 = \mathcal{O}(1/n)$, the first term is $\mathcal{O}(1/n)$. For the case $\mathbb{E}[\epsilon\epsilon^T] = I$, we have $\text{Tr}(\mathbb{E}[\epsilon\epsilon^\top]P_{k_2}) = k_2$. As $\lambda_2 \to \infty$, $\Delta_{\text{stat}} \to 0$, indicating faster statistical convergence with stronger shrinkage. Last, $\Delta_{\text{bias}}$ is the regularization-induced bias. As $\lambda_2 \to 0$, $P_{k_2} \to \mathcal{P}_{k_2}^r(\Sigma^{1/2}\Delta_2^T)$, and the first term becomes $\sum_{i>k_2} \sigma_i^2(\Delta_2\Sigma^{1/2})$. When $\Delta_2$ is low-rank with $r_2 \leq k_2$, this term vanishes. The second term, due to $\lambda_2$-shrinkage, vanishes as $\lambda_2 \to 0$. This decomposition highlights how $\lambda_2$ and $k_2$ trade off statistical efficiency against model bias. We next analyze the consistency of estimating $\Delta_1$ and $\Delta_2$.

**Theorem 4.2** (Estimation Consistency of $\Delta_1$). *Assume the data-generating model (1) and Assumptions 4.1 - 4.3 hold. Then,*

$$E\|\widehat{\Delta}_1 - \Delta_1\|_F \lesssim \Delta_{stat} + \Delta_{bias},$$

*where $\Delta_{stat} = \frac{\sqrt{k_1}}{\sqrt{n}(1+\lambda_1)}$ and $\Delta_{bias} = \frac{\delta}{1+\lambda_1} + \frac{\lambda_1\|\Delta_1\|_F}{1+\lambda_1} + \frac{1}{1+\lambda_1}\sqrt{\sum_{i>k_1}\sigma^2\left(W_1(VV^T - I)\right)}.$*

The statistical error $\Delta_{\text{stat}}$ decays as $\mathcal{O}(1/\sqrt{n})$, with constants depending on $\lambda_1$ and $k_1$. Stronger shrinkage (large $\lambda_1$) or lower rank (small $k_1$) accelerates convergence. The bias $\Delta_{\text{bias}}$ has three components. The first term measures the deviation $\delta = \|\Delta_1 VV^T\|_F$ from the projection assumption. The second term is the shrinkage bias. As $\lambda_1 \to \infty$, $\widehat{\Delta}_1 \to 0$, and the bias approaches $\|\Delta_1\|_F$. The last term is the rank constraint bias, which vanishes if $\text{rank}(W_1(VV^\top - I)) \leq k_1$.

When the projection adaptation assumption holds exactly, we obtain the following result:

**Corollary 1.** *Suppose there exists a projection matrix $P$ and $\Delta_2 \in \mathbb{R}^{q \times l_2}$ such that*

$$Y_i = (W_2 + \Delta_2)a\left(W_1 P\phi(X_i)\right) + \epsilon_i \tag{14}$$

*Then, with $k_1 = \min\{l_1, l_2\}$ and $\lambda_1 = 0$, the estimator $\widehat{\Delta}_1 = -W_1(I - \widehat{V}\widehat{V}^\top)$ is consistent. That is, $\mathbb{E}\|\widehat{\Delta}_1 - \Delta_1\|_F \to 0$ as $n \to \infty$.*

The corollary follows directly from $\delta = 0$ and Theorem 4.2. We now turn to consistency of $\Delta_2$:

**Theorem 4.3** (Estimation Consistency of $\Delta_2$). *Assume the data-generating model (1) and Assumptions 4.1 - 4.3 hold. Then,*

$$\mathbb{E}\|\widehat{\Delta}_2 - \Delta_2\|_F \lesssim \Delta_{bias} + \Delta_{stat} + \Delta_{first}, \tag{15}$$

*where*

$$\Delta_{bias} = \|(I - P_k)\Delta_2\|_F + \mathbb{E}\|P_k\Delta_2\left(I - \Sigma_n(\Sigma_n + \lambda_2 I)^{-1}\right)\|_F,$$

$$\Delta_{stat} = \left(\frac{K_{\lambda_2}}{\lambda_2}\sqrt{k_2} + \frac{1}{\lambda_2}\sqrt{Tr(\mathbb{E}[\epsilon_i\epsilon_i^T]P_{k_2})}\right)\frac{1}{\sqrt{n}},$$

$$\Delta_{first} = \left(\frac{K_{\lambda_2}}{\lambda_2} + C_{\lambda_2}\right)\mathbb{E}\|\Delta_1 - \widehat{\Delta}_1\|_{op}.$$

*Here, $C_{\lambda_2}$, $K_{\lambda_2}$, and $P_{k_2}$ are as defined in Theorem 4.1, and $\Sigma_n$ is the empirical covariance of $a((W_1 + \widehat{\Delta}_1)\phi(X))$.*

Interpretation of Theorem 4.3 is similar to Theorem 4.1. The statistical error $\Delta_{\text{stat}}$ decays faster under stronger shrinkage (large $\lambda_2$) or lower rank (small $k_2$). The error due to first layer estimation $\Delta_{\text{first}}$ vanishes as $\lambda_2 \to \infty$, since $\widehat{\Delta}_2 \to 0$ independently of $\widehat{\Delta}_1$. The bias $\Delta_{\text{bias}}$ contains a rank constraint term (less interpretable than in Theorem 4.1, since $P_{k_2}$ does not directly project $\Delta_2$) and a shrinkage term vanishing as $\lambda_2 \to 0$.

Combining Theorems 4.1, 4.2, and 4.3, we conclude that when the projection adaptation condition (14) holds, our estimators $(\widehat{\Delta}_1, \widehat{\Delta}_2)$ converge to the true parameters $(\Delta_1, \Delta_2)$ as $n \to \infty$. Consequently, in the absence of regularization, the excess risk vanishes asymptotically. This establishes the theoretical validity of our two-step adaptation framework.

## 5 NUMERICAL EXPERIMENTS

### 5.1 SIMULATIONS

We conduct simulation experiments under the projection adaptation assumption (Assumption 2.1). Specifically, we generate responses according to $Y = (W_2 + \Delta_2)a(W_1 PX) + \epsilon$, where $P$ is a projection matrix, $a(\cdot)$ is the ReLU activation function, and $\epsilon \sim \mathcal{N}(0, I_{100})$ is the Gaussian noise. This model suggests that only a lower-dimensional subspace of the input $X$ is relevant to the target.

We consider a two-layer network with input dimension $l_1 = 200$, hidden dimension $l_2 = 100$, and output dimension $q = 50$. Inputs $X$ are drawn i.i.d. from $X \sim \mathcal{N}(0, I_{200})$, mimicking standardized feature representations obtained from a pretrained regularized encoder. The weight matrices $(W_1, W_2, \Delta_2)$, are generated with i.i.d. entries from $\mathcal{N}(0, \sigma^2 = 0.25^2)$. While for $\Delta_1$, it is generated according to $\Delta_1 = W_1 P - W_1$, where $P \in \mathbb{R}^{l_1 \times l_1}$ has rank 75.

We compare the proposed approach with two baselines. The first is the reduced rank ridge (denoted as RRR) regression, which estimates only $\Delta_2$ while keeping the first layer fixed. This baseline isolates the benefit of adapting the first layer via subspace projection. The second baseline implements LoRA (Hu et al., 2022) using Stochastic Gradient descent (SGD). In this setup, each $\Delta_i$ is factored as trainable parameters $A_i B_i^T$ with $A_i \in \mathbb{R}^{l_{i+1} \times k}, B_i \in \mathbb{R}^{l_i \times k}$. We train the parameters with a learning rate of $0.001$, batch size 256, minimizing the mean squared error of a training dataset with sample size $n$. The performance of the prior model (i.e., $\Delta_1 = \Delta_2 = 0$) is reported as PT.

Results on a test dataset of sample size 1000 are reported in Table 1. From the result, all three methods improve the pre-trained models by lowering its MSE over target task. Our proposed method is comparable to SGD-based LoRA which has been trained for 30 epochs over the train set. Moreover, it uniformly outperforms the baseline RRR across all sample size $n$ and rank constraint $k$, suggesting that there is value in updating the first layer parameters. Another observation is that our method performs better than SGD in low sample size, low rank setting (e.g., $n$ from 5000 to 30000, $k = 5, 7$). Finally, we shall note that our method computes in seconds, significantly faster than SGD-based LoRA.

| | | | k=5 | | | k=7 | | | k=10 | |
| n | PT | RRR | Proposed | SGD | RRR | Proposed | SGD | RRR | Proposed | SGD |
|---|---|---|---|---|---|---|---|---|---|---|
| 5000 | 36.474 | 27.071 | 25.303 | 27.328 | 29.304 | 27.840 | 29.813 | 27.071 | 25.303 | 27.328 |
| 10000 | 36.617 | 26.810 | 24.618 | 26.368 | 29.247 | 27.398 | 29.339 | 26.810 | 24.618 | 26.368 |
| 15000 | 36.252 | 26.717 | 24.417 | 25.538 | 29.120 | 27.112 | 28.360 | 26.717 | 24.417 | 25.538 |
| 25000 | 36.900 | 26.958 | 24.188 | 24.300 | 29.160 | 26.961 | 27.695 | 26.958 | 24.188 | 24.300 |
| 30000 | 36.630 | 26.650 | 23.958 | 23.587 | 29.006 | 26.724 | 26.788 | 26.650 | 23.958 | 23.587 |
| 40000 | 36.385 | 26.677 | 23.744 | 22.685 | 29.026 | 26.614 | 25.709 | 26.677 | 23.744 | 22.685 |
| 50000 | 36.775 | 26.683 | 23.694 | 22.252 | 29.144 | 26.576 | 25.077 | 26.683 | 23.694 | 22.252 |
| 100000 | 36.374 | 26.652 | 22.945 | 21.044 | 28.982 | 26.085 | 23.422 | 26.652 | 22.945 | 21.044 |

**Table 1:** Comparison of MSE under different rank $k$ and training sizes ($n$).

## 5.2 MNIST IMAGE INPAINTING

We evaluate the performance of our estimators on the MNIST image inpainting task. Each image is divided into two parts: the lower-left $14 \times 14$ region serves as the response $Y \in \mathbb{R}^{196}$, and the remaining 756 pixels form the predictor $X \in \mathbb{R}^{756}$. The digit class label is denoted by $C \in \{0, \ldots, 9\}$.

We consider a transfer learning setup where a model is pre-trained to predict $Y$ from $X$ using data from all digit classes except a held-out target class $c \in \{0, \ldots, 9\}$. The goal is to adapt the pre-trained model to perform inpainting for class $c$, despite having never observed any examples from this class during pre-training.

To obtain a latent representation, we first train a Wasserstein Autoencoder (WAE) (Tolstikhin et al., 2017), denoted $\phi \colon \mathbb{R}^{756} \to \mathbb{R}^{20}$, to encode $X$ into a 20-dimensional latent code $Z = \phi(X)$. During training, the WAE regularizes the marginal distribution of $Z$ to approximate $\mathcal{N}(0, I_{20})$. For the target class $c$, we estimate the conditional mean $\mu_c = \mathbb{E}[Z \mid C = c]$ and covariance $\Sigma_c = \mathrm{Cov}(Z \mid C = c)$ from target training samples. Using Stein's lemma, we approximate the second-order score function of $Z \mid C = c$ as

$$S(z) \approx \Sigma_c^{-1}(z - \mu_c)(z - \mu_c)^\top \Sigma_c^{-1}.$$

We assume the following generative models:

1. For digits $\{0, \ldots, 9\} \setminus \{c\}$: $Y = W_2 a(W_1 \phi(X)) + \epsilon$,
2. For digit $c$: $Y = (W_2 + \Delta W_2) a((W_1 + \Delta W_1) \phi(X)) + \epsilon$,

where $a(\cdot)$ is ReLU activation function and $\epsilon$ is additive noise. The weight matrices have dimension $l_1 = 20, l_2 = 1024, q = 196$. We aim to adapt both layers of the network via low-rank matrices $(\Delta_1, \Delta_2)$.

As in the simulation study, we focus on full adaptation and thus fix the shrinkage penalties at $\lambda_1 = \lambda_2 = 1$ to avoid bias toward the pre-trained weights. Instead, we vary the shared rank constraint $k$, enforcing $\mathrm{rank}(\Delta W_1) = \mathrm{rank}(\Delta W_2) \leq k$, and examine its impact on estimation accuracy.

For comparison, we include two baselines. The first, LoRA (Hu et al., 2022), factorizes each $\Delta_i = A_i B_i^T$ with trainable matrices $A_i \in \mathbb{R}^{l_{i+1} \times k}$, $B_i \in \mathbb{R}^{l_i \times k}$, initialized as $A_i \sim \mathcal{N}(0, I)$, $B_i = 0$. We train using SGD with learning rate 0.01, batch size 32, for 100 epochs on a dataset of size $n = 48,000$, minimizing mean squared error. The second baseline, SGD-I, uses the same LoRA parameterization but initializes $A_i B_i^\top = 10e^{-3} \cdot \widehat{\Delta}_i$, where $\widehat{\Delta}_i$ is our proposed estimator. This tests whether our closed-form estimate provides a better initialization than zero initialization. The performance of prior model (i.e., $\Delta_1 = \Delta_2 = 0$) is reported as PT.

We evaluate our method across all target classes $c \in \{0, \ldots, 9\}$ and report test performance on a test set of size $10,000$ in Tables 2. As shown in Table 2, our closed-form estimator significantly reduces prediction error on unseen target classes compared to the pre-trained baseline. Notably, it outperforms SGD-based LoRA trained for 100 epochs in the majority of classes — despite using standard, untuned hyperparameters (learning rate, batch size) for SGD. This suggests that our method is not only computationally efficient but also robust to hyperparameter sensitivity, a common pitfall of iterative optimization. Moreover, our method updates in seconds, significantly faster than SGD which takes minutes.

Furthermore, initializing SGD-based LoRA with our estimator (SGD-I) consistently improves final performance over the conventional zero initialization (SGD). This confirms that our analytical solution provides a high-quality, data-adaptive starting point that accelerates convergence and enhances final accuracy. This highlights its value both as a standalone adaptation tool and as a warm-start mechanism for iterative fine-tuning.

| target | PT | k=1 | | | k=2 | | | k=3 | | |
|--------|------|----------|--------|--------|----------|--------|--------|----------|--------|--------|
| | | Proposed | SGD-I | SGD | Proposed | SGD-I | SGD | Proposed | SGD-I | SGD |
| c=0 | 0.1125 | 0.0714 | 0.0696 | 0.0722 | 0.0680 | 0.0669 | 0.0701 | 0.0672 | 0.0660 | 0.0707 |
| c=1 | 0.0595 | 0.0321 | 0.0340 | 0.0368 | 0.0275 | 0.0305 | 0.0368 | 0.0262 | 0.0274 | 0.0357 |
| c=2 | 0.0993 | 0.0772 | 0.0773 | 0.0780 | 0.0758 | 0.0763 | 0.0784 | 0.0756 | 0.0759 | 0.0774 |
| c=3 | 0.0831 | 0.0582 | 0.0587 | 0.0596 | 0.0593 | 0.0585 | 0.0596 | 0.0586 | 0.0585 | 0.0596 |
| c=4 | 0.0713 | 0.0512 | 0.0526 | 0.0529 | 0.0497 | 0.0524 | 0.0528 | 0.0501 | 0.0522 | 0.0529 |
| c=5 | 0.0773 | 0.0667 | 0.0662 | 0.0688 | 0.0650 | 0.0653 | 0.0686 | 0.0646 | 0.0651 | 0.0678 |
| c=6 | 0.0875 | 0.0540 | 0.0549 | 0.0559 | 0.0506 | 0.0531 | 0.0560 | 0.0500 | 0.0527 | 0.0560 |
| c=7 | 0.0669 | 0.0428 | 0.0435 | 0.0451 | 0.0428 | 0.0426 | 0.0448 | 0.0417 | 0.0425 | 0.0450 |
| c=8 | 0.0788 | 0.0624 | 0.0632 | 0.0634 | 0.0626 | 0.0626 | 0.0634 | 0.0635 | 0.0625 | 0.0635 |
| c=9 | 0.0562 | 0.0485 | 0.0485 | 0.0497 | 0.0480 | 0.0482 | 0.0496 | 0.0476 | 0.0481 | 0.0497 |

**Table 2:** Comparison of MSE across update strategies.

# 6 DISCUSSIONS

We proposed a closed-form, non-iterative approach for solving LoRA in two-layer networks, establishing its statistical consistency under a projection adaptation assumption — that optimal adaptation acts as a subspace projection removing irrelevant feature directions. Our method achieves accuracy comparable to SGD while adapting in seconds, and serves as a good initialization for iterative training algorithms. The approach can extend to deeper feedforward networks provided second-order score functions for hidden neuron activations are available, enabling layer-wise analytical adaptation without backpropagation. Its speed make it particularly suited for real-time transfer settings such as on-device personalization or edge federated learning. While the projection assumption is theoretically tractable and empirically plausible in semantically aligned tasks, its broader validity warrants further investigation.

## REPRODUCIBILITY STATEMENT

We have performed the following actions to ensure reproducibility of our work. All experiments use fixed random seeds to ensure deterministic outcomes. The code to reproduce our results are provided as supplementary material. The MNIST dataset used in our experiment is publicly available at `https://archive.ics.uci.edu/dataset/683/mnist+database+of+handwritten+digits`. Proofs of theoretical results are included in full in the appendix.

## ETHICS STATEMENT

All authors acknowledge that they have read and adhered to the ICLR Code of Ethics.

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

# A  APPENDIX

## USE OF LARGE LANGUAGE MODELS

We have used large language models (LLMs) as a writing assistance tool during the preparation of this paper. The LLMs were employed to improve grammar, correct typographical errors, and enhance sentence clarity and structure. However, they were not involved in generating research ideas, experimental design or data analysis.

## A.1  PROOF OF THEOREMS

**Notations:** For a matrix $A \in \mathbb{R}^{m \times n}$, we use $\sigma_i(A)$ to denote its $i$th largest singular value (i.e, $\sigma_1(A) \geq \sigma_2(A) \geq \dots$), and $\lambda_i(A)$ to denote its $i$th largest eigenvalue. We use $(A)_{i,\cdot}$ to denote the $i$-th row of $A$, and $(A)_{\cdot,i}$ to denote the $i$-th column of $A$. We define the unit sphere as $\mathcal{S}^{n-1} := \{x \in \mathbb{R}^n : \|x\|_2 = 1\} \subset \mathbb{R}^n$. We use $\mathrm{Col}(A)$ to denote the column space of a matrix $A$, and $\mathrm{Row}(A)$ to denote its row space. Suppose $A \in \mathbb{R}^{n \times m}$ is a rank $r_A$ matrix with Singular Value Decomposition (SVD) of the form $A = USV^T$, where $U = \begin{bmatrix} U_k, U_k^\perp \end{bmatrix} \in \mathbb{R}^{n \times r_A}, U_k \in \mathbb{R}^{n \times k}$, $S \in \mathbb{R}^{r_A \times r_A}$ is diagonal, $V = \begin{bmatrix} V_k, V_k^\perp \end{bmatrix} \in \mathbb{R}^{m \times r_A}$, and $V_k \in \mathbb{R}^{m \times k}$. We write $\mathrm{SVD}_r(A) = V$ and $\mathrm{SVD}_l(A) = U$. Moreover, we use $\mathcal{P}_k^r(A) = V_k V_k^T$ to denote the orthogonal projection matrix onto the subspace spanned by its top-$k$ right singular vectors, $\mathcal{P}_k^l(A) = U_k U_k^T$ for the top-$k$ left

singular vectors, and $\mathcal{R}_k(A) = [U_k, 0]S[V_k, 0]^T$ to denote its rank-$k$ approximation. We use $\|A\|_F$ to denote its Frobenius norm, $\|A\|_{op}$ to denote its operator norm, and $\text{rank}(A)$ to denote its rank. For a vector $v \in \mathbb{R}^n$, we use $\|v\|_2$ to denote its $\ell_2$-norm. We say that $A_n \lesssim B_n$ (also written as $A = O(B_n)$) if there exists $C > 0$ such that $A \leq CB_n$.

### A.1.1 LEMMA A.1.1

Let $X \in \mathbb{R}^{n \times p}$, $Y \in \mathbb{R}^{n \times k}$, $B \in \mathbb{R}^{p \times k}$ are three matrices, and $\lambda > 0$ be a constant. Consider the optimization problem:

$$\min_{\{B:\text{rank}(B) \leq k\}} \|Y - XB\|_F^2 + \lambda\|B\|_F^2.$$

The solution is $\widehat{B}_R^* \, \mathcal{P}_k^r\left(Y^T X \widehat{B}_R^*\right)$, where $\widehat{B}_R^* = \left(X^T X + \lambda I\right)^{-1} X^T Y$.

**Proof:** This is the result in Mukherjee & Zhu (2011).

### A.1.2 PROPOSITION 3.2

For simplicity, we write $V = \widehat{V}$. $X = I - VV^T$ is a projection matrix which satisfies $XX = X$, $X^T = X$. Let $U$ be a matrix whose columns form an orthonormal basis for the orthogonal complement of $\text{Col}(V)$, and it satisfies $U^T V = V^T U = 0$. By the uniqueness of orthogonal projection matrix, $X = UU^T$ and $I = VV^T + UU^T$.

$$X + \lambda I = UU^T + \lambda VV^T + \lambda UU^T$$
$$= (1 + \lambda)UU^T + \lambda VV^T$$

The inverse is $(X + \lambda I)^{-1} = \frac{1}{\lambda}VV^T + \frac{1}{1+\lambda}UU^T$. Apply Lemma A.1.1 with $X = (I - VV^T)$, $Y = -(I - VV^T)W_1^T$, and $B = \Delta_1^T$. The solution to the optimization problem (3.2) is

$$\widehat{\Delta}_1^T = \left(X^T X + \lambda I\right)^{-1} X^T Y \mathcal{P}_{k_1}^r\left(Y^T X \left(X^T X + \lambda I\right)^{-1} X^T Y\right)$$

Simplifying the terms give

$$\widehat{\Delta}_1 = \frac{1}{1 + \lambda_1}\mathcal{R}_{k_1}\left(-W_1(I - VV^T)\right).$$

### A.1.3 PROPOSITION 3.3

Apply Lemma A.1.1 with target matrix $Y - \widehat{A}W_2^T$, we obtain the optimizer.

### A.1.4 LEMMA A.1.4

Let $X_1, \ldots, X_n$ be independent, mean zero, $p \times p$ symmetric random matrices, such that $\|X_i\|_{op} \leq K$ almost surely for all $i$. Then, for every $t \geq 0$, we have

$$\mathbb{P}\left(\|\sum_{i=1}^n X_i\|_{op} \geq t\right) \leq 2p \exp\left(-\frac{t^2/2}{\sigma^2 + Kt/3}\right),$$

where $\sigma^2 = \|\sum_{i=1}^n \mathbb{E}X_i^2\|_{op}$ is the operator norm of the matrix variance of the sum.

**Proof:** This is Theorem 5.4.1 in Vershynin (2018)

### A.1.5 LEMMA A.1.5

Suppose model (1), Assumption 4.1 - 4.2 hold. Define $E = \frac{1}{n}\sum_{i=1}^n Y_{ij}S(Z_i) - \mathbb{E}[Y_{ij}S(Z_i)]$. Then, there exists a constant $C > 0$ such that $\|E\|_{op} \leq C$. And, for any $t \geq 0$,

$$\mathbb{P}\left(\|E\|_{op} \geq t\right) \leq 2l_1 \exp\left(\frac{-nt^2/2}{C^2 + Ct/3}\right). \tag{16}$$

**Proof:** Define $E_i = \frac{1}{n}Y_{ij}S(Z_i) - \frac{1}{n}\mathbb{E}[Y_{ij}S(Z_i)]$, then $E_1, \ldots, E_n$ are independent, mean zero, $l_1 \times l_1$ symmetric random matrices. By assumption 4.2, $\|S(Z_i)\|_{op} \leq c_3$ and

$$
\begin{aligned}
|Y_{ij}| &= |(W_2 + \Delta_2)_{j,.} a((W_1 + \Delta_1)Z_i) + \epsilon_{ij}| \\
&\leq K\|(W_2 + \Delta_2)_{j,.}\|_2 \|W_1 + \Delta_1\|_{op}\|Z_i\|_2 + |\epsilon_{ij}| \\
&\leq K\|(W_2 + \Delta_2)_{j,.}\|_2 \|W_1 + \Delta_1\|_{op}c_1 + c_2.
\end{aligned}
$$

Thus, there exists a constant $c_4$ such that $|Y_{ij}| \leq c_4$ almost surely. Consequently,

$$
\begin{aligned}
\|E_i\|_{op} &\leq \frac{1}{n}\|Y_{ij}S(Z_i)\|_{op} + \frac{1}{n}\|\mathbb{E}[Y_{ij}S(Z_i)]\|_{op} \\
&\leq \frac{2c_3c_4}{n}.
\end{aligned}
$$

Let $C = 2c_3c_4$, and $E = \sum_{i=1}^n E_i$, we have $\sigma^2 = \|\sum_{i=1}^n \mathbb{E}E_i^2\|_{op} \leq \sum_{i=1}^n \|\mathbb{E}E_i^2\|_{op} \leq n\frac{4c_3^2c_4^2}{n^2} = \frac{C^2}{n}$. By Lemma A.1.4,

$$
\mathbb{P}\left(\|E\|_{op} \geq t\right) \leq 2l_1 \exp\left(\frac{-nt^2/2}{C^2 + Ct/3}\right).
$$

### A.1.6   LEMMA A.1.6

Let $\Sigma, \hat{\Sigma} \in \mathbb{R}^{p \times p}$ be symmetric, with eigenvalues $\lambda_1 \geq \ldots \geq \lambda_p$ and $\hat{\lambda}_1 \geq \ldots \geq \hat{\lambda}_p$ respectively. Fix $1 \leq r \leq s \leq p$ and assume that $\min(\lambda_{r-1} - \lambda_r, \lambda_s - \lambda_{s+1}) > 0$, where $\lambda_0 := \infty$ and $\lambda_{p+1} := -\infty$. Let $d := s - r + 1$, and let $V = (v_r, v_{r+1}, \ldots, v_s) \in \mathbb{R}^{p \times d}$ and $\hat{V} = (\hat{v}_r, \hat{v}_{r+1}, \ldots, \hat{v}_s) \in \mathbb{R}^{p \times d}$ have orthonormal columns satisfying $\Sigma v_j = \lambda_j v_j$ and $\hat{\Sigma}\hat{v}_j = \hat{\lambda}_j \hat{v}_j$ for $j = r, r+1, \ldots, s$. Then

$$
\|\sin\Theta(\hat{V}, V)\|_{\mathrm{F}} \leq \frac{2\min(d^{1/2}\|\hat{\Sigma} - \Sigma\|_{\mathrm{op}}, \|\hat{\Sigma} - \Sigma\|_{\mathrm{F}})}{\min(\lambda_{r-1} - \lambda_r, \lambda_s - \lambda_{s+1})}. \tag{17}
$$

Moreover, there exists an orthogonal matrix $\hat{O} \in \mathbb{R}^{d \times d}$ such that

$$
\|\hat{V}\hat{O} - V\|_{\mathrm{F}} \leq \frac{2^{3/2}\min(d^{1/2}\|\hat{\Sigma} - \Sigma\|_{\mathrm{op}}, \|\hat{\Sigma} - \Sigma\|_{\mathrm{F}})}{\min(\lambda_{r-1} - \lambda_r, \lambda_s - \lambda_{s+1})}. \tag{18}
$$

**Proof:** This is Theorem 2 in Yu et al. (2015)

### A.1.7   LEMMA: A.1.7

Let $A, \hat{A} \in \mathbb{R}^{p \times q}$ have singular values $\sigma_1 \geq \ldots \geq \sigma_{\min(p,q)}$ and $\hat{\sigma}_1 \geq \ldots \geq \hat{\sigma}_{\min(p,q)}$ respectively. Fix $1 \leq r \leq s \leq \mathrm{rank}(A)$ and assume that $\min(\sigma_{r-1}^2 - \sigma_r^2, \sigma_s^2 - \sigma_{s+1}^2) > 0$, where $\sigma_0^2 := \infty$ and $\sigma_{q+1}^2 := -\infty$. Let $d := s - r + 1$, and let $V = (v_r, v_{r+1}, \ldots, v_s) \in \mathbb{R}^{q \times d}$ and $\hat{V} = (\hat{v}_r, \hat{v}_{r+1}, \ldots, \hat{v}_s) \in \mathbb{R}^{q \times d}$ have orthonormal columns satisfying $Av_j = \sigma_j u_j$ and $\hat{A}\hat{v}_j = \hat{\sigma}_j \hat{u}_j$ for $j = r, r+1, \ldots, s$. Then, there exists an orthogonal matrix $\hat{O} \in \mathbb{R}^{d \times d}$ such that

$$
\|\hat{V}\hat{O} - V\|_{\mathrm{F}} \leq \frac{2^{3/2}(2\sigma_1 + \|\hat{A} - A\|_{\mathrm{op}})d^{1/2}\|\hat{A} - A\|_{\mathrm{op}}}{\min(\sigma_{r-1}^2 - \sigma_r^2, \sigma_s^2 - \sigma_{s+1}^2)}.
$$

**Proof:** $A^T A \in \mathbb{R}^{q \times q}$ is a symmetric matrix with eigenvalues $\sigma_1^2 \geq \cdots \geq \sigma_q^2$, and $\hat{A}^T\hat{A} \in \mathbb{R}^{q \times q}$ is a symmetric matrix with eigenvalues $\hat{\sigma}_1^2 \geq \cdots \geq \hat{\sigma}_q^2$. Moreover, $A^T A v_j = \sigma_j^2 v_j$ and $\hat{A}^T\hat{A}\hat{v}_j = \hat{\sigma}_j^2\hat{v}_j$ for $j = r, r+1, \ldots, s$. By Lemma A.1.6, there exists an orthogonal matrix $\hat{O} \in \mathbb{R}^{d \times d}$ such that

$$
\|\hat{V}\hat{O} - V\|_{\mathrm{F}} \leq \frac{2^{3/2}d^{1/2}\|\hat{A}^T\hat{A} - A^T A\|_{\mathrm{op}}}{\min(\sigma_{r-1}^2 - \sigma_r^2, \sigma_s^2 - \sigma_{s+1}^2)}.
$$

And finally,

$$
\begin{aligned}
\|\hat{A}^T\hat{A} - A^T A v_j\|_{\mathrm{op}} &= \|(\hat{A} - A)^T\hat{A} - A^T(A - \hat{A})\|_{op} \\
&\leq \left(\|\hat{A}\|_{op} + \|A\|_{op}\right)\|\hat{A} - A\|_{op} \\
&\leq \left(2\sigma_1 + \|\hat{A} - A\|_{op}\right)\|\hat{A} - A\|_{op}
\end{aligned}
$$

### A.1.8   LEMMA A.1.8

Let $A, \widehat{A} \in \mathbb{R}^{p \times q}$ be two matrices, $V$ be a matrix whose columns are the top $k$ right singular values of $A$, and $\widehat{V}$ be that of $\widehat{A}$. Consider the orthogonal projection matrices $P = VV^T$ and $\widehat{P} = \widehat{V}\widehat{V}^T$. For any orthogonal matrix $R$,

$$\|\widehat{P} - P\|_F \leq 4\|\widehat{V}R - V\|_F$$

**Proof:** First, $\|\widehat{V}R - V\|_{op} \leq \|V\|_{op} + \|\widehat{V}R\|_{op} = 2$. For any orthogonal matrix $R$,

$$
\begin{aligned}
\|\widehat{P} - P\|_F &= \|\widehat{V}\widehat{V}^T - VV^T\|_F \\
&= \|(\widehat{V}R - V)(\widehat{V}R - V)^T + (\widehat{V}R - V)V^T + V(\widehat{V}R - V)^T\|_F \\
&\leq \|(\widehat{V}R - V)(\widehat{V}R - V)^T\|_F + \|(\widehat{V}R - V)V^T\|_F + \|V(\widehat{V}R - V)^T\|_F \\
&\leq \|\widehat{V}R - V\|_F \left( \|\widehat{V}R - V\|_{op} + 2\|V\|_{op} \right) \\
&\leq 4\|\widehat{V}R - V\|_F
\end{aligned}
$$

### A.1.9   LEMMA A.1.9

Let $A, \widehat{A} \in \mathbb{R}^{p \times q}$ be two matrices, $V$ be a matrix whose columns are the top $k$ right singular values of $A$, and $\widehat{V}$ be that of $\widehat{A}$. Consider the orthogonal projection matrices $P = VV^T$ and $\widehat{P} = \widehat{V}\widehat{V}^T$. If there exists a constant $C$ such that $\|A - \widehat{A}\|_{op} < C$, then,

$$\|\widehat{P} - P\|_F \lesssim \sqrt{k}\|A - \widehat{A}\|_{op}$$

**Proof:** By Lemma A.1.8, for any orthogonal matrix $R$,

$$\|\widehat{P} - P\|_F \leq 4\|\widehat{V}R - V\|_F$$

Let $\sigma_1 = \sigma_1(A)$, $\sigma_k = \sigma_k(A)$ and $\sigma_{k+1} = \sigma_{k+1}(A)$. By Lemma A.1.7, there exists an orthogonal matrix $R$ such that

$$
\begin{aligned}
\|\widehat{V}R - V\|_F &\leq \frac{2^{2/3}(2\sigma_1 + \|A - \widehat{A}\|_{op})\sqrt{k}\|A - \widehat{A}\|_{op}}{\sigma_k^2 - \sigma_{k+1}^2} \\
&\leq \frac{2^{2/3}(2\sigma_1 + C)\sqrt{k}\|A - \widehat{A}\|_{op}}{\sigma_k^2 - \sigma_{k+1}^2} \\
&\lesssim \sqrt{k}\|A - \widehat{A}\|_{op}
\end{aligned}
$$

### A.1.10   LEMMA A.1.10

Let $X \in \mathbb{R}^{n \times p}, E \in \mathbb{R}^{n \times k}, B^* \in \mathbb{R}^{p \times k}$ and $x_i^T$ and $e_i^T$ be the rows of $X$ and $E$ respectively. Assume there exists constants $c_1, c_2$ such that $\|x_i\|_2 \leq c_1, \|e_i\|_2 \leq c_2, \forall i \in [n]$. For a positive constant $\lambda > 0$, define $\widehat{\Sigma}_n := \frac{1}{n}X^TX$ and $\widehat{B}_\lambda := \frac{1}{n}(\widehat{\Sigma}_n + \lambda I)^{-1}X^T(XB^* + E)$. Then,

1. $\|\widehat{\Sigma}_n\|_{op} \leq c_1^2$

2. $\frac{1}{n}\|X^TE\|_F \leq c_1 c_2$

3. If $x_i$ and $e_i$ are mutually independent, and $\mathbb{E}e_i = 0$, then $\mathbb{E}\|\frac{X^TE}{n}\|_F^2 \leq \frac{c_1^2 c_2^2}{n}$ and $\mathbb{E}\|\frac{X^TE}{n}\|_F \leq \frac{c_1 c_2}{\sqrt{n}}$.

4. $\|\widehat{B}_\lambda\|_F \leq \frac{1}{\lambda}c_1(\|B^*\|_{op}c_1 + c_2) \lesssim \frac{1}{\lambda}(\|B^*\|_{op} + 1)$.

**Proof:**

1. $\|\widehat{\Sigma}\|_{op} = \frac{1}{n}\|\sum_{i=1}^n x_i x_i^T\|_{op} \leq \frac{1}{n}\sum_{i=1}^n \|x_i x_i^T\|_{op} \leq \frac{1}{n}nc_1^2 = c_1^2$

2. $\frac{1}{n}\|X^TE\|_F = \frac{1}{n}\|\sum_{i=1}^n x_i e_i^T\|_F \leq \frac{1}{n}\sum_{i=1}^n \|x_i e_i^T\|_F = \frac{1}{n}\sum_{i=1}^n \|x_i\|_2\|e_i\|_2 \leq c_1 c_2$

3. $\mathbb{E}\|\frac{X^T E}{n}\|_F = \mathbb{E}\sqrt{\|\frac{X^T E}{n}\|_F^2} \leq \sqrt{\mathbb{E}\|\frac{X^T E}{n}\|_F^2}$ . And, $\mathbb{E}\|\frac{X^T E}{n}\|_F^2 = \frac{1}{n^2}\mathbb{E}Tr(\sum_{i=1}^n e_i x_i^T x_i e_i^T + \sum_{i \neq j} e_i x_i^T x_j e_j^T) \leq \frac{1}{n^2}\sum_{i=1}^n \mathbb{E}\|x_i\|_2^2 \mathbb{E}\|e_i\|_2^2 + 0 \leq \frac{c_1^2 c_2^2}{n}$.

4. Apply 2 to $\|\frac{1}{n}X^T(XB^* + E)\|_F$, we have

$$\|\widehat{B}_\lambda\|_F \leq \|(\widehat{\Sigma}_n + \lambda I)^{-1}\|_{op}\|\frac{1}{n}X^T(XB^* + E)\|_F$$

$$\leq \frac{1}{\lambda}c_1(\|B^*\|_{op}c_1 + c_2).$$

### A.1.11 LEMMA: A.1.11

Let $X, D \in \mathbb{R}^{n \times p}$ and $Y, F \in \mathbb{R}^{n \times q}$ be four matrices whose rows are $x_i, d_i, y_i, f_i$, respectively. Let $\lambda > 0$ be a positive constant. Suppose there exists a constant $c > 0$ such that $\|x_i\|_2, \|d_i\|_2, \|y_i\|_2, \|f_i\|_2 \leq c$, for all $i \in [n]$. Then, for the function

$$B(t) = \left\{ \frac{(X + tD)^T(X + tD)}{n} + \lambda I \right\}^{-1} \frac{1}{n}(X + tD)^T(Y + tf). \tag{19}$$

We have

$$B'(0) = \left( \frac{X^T X}{n} + \lambda I \right)^{-1} \left( \frac{X^T f}{n} - \frac{X^T D\, B(0)}{n} + \frac{D^T(Y - X\, B(0))}{n} \right)$$

And, for any fixed constant $k > 0$, there exists a constant $C_k$ such that $\|B''(t)\|_{op} \leq C_k \left( \frac{1}{\lambda} + \frac{1}{\lambda^2} + \frac{1}{\lambda^3} \right)$ for all $t \in [0, k]$.

**Proof:** Let $A(t) = \left( \frac{(X+tD)^T(X+tD)}{n} + \lambda I \right)$ and $b(t) = \frac{(X+tD)^T(Y+tf)}{n}$. Then, $B(t) = A^{-1}(t)b(t)$. Apply chain rule with $\frac{d}{dt}\left(A(t)^{-1}\right) = -A(t)^{-1}A'(t)A(t)^{-1}$, we have $B'(t) = (-A^{-1}(t)A'(t)A^{-1}(t))b(t) + A^{-1}(t)b'(t)$. Since $A^{-1}(t)b(t) = B(t)$, it simplifies to $B'(t) = A^{-1}(t)[b'(t) - A'(t)B(t)]$.

Similarly taking derivatives, we have $b'(t) = \frac{D^T(Y+tf)+(X+tD)^T f}{n}$ and $A'(t) = \frac{D^T(X+tD)+(X+tD)^T D}{n}$. Define $g(t) = b'(t) - A'(t)B(t)$. Apply chain rule again, $B''(t) = -A^{-1}(t)A'(t)A^{-1}(t)g(t) + A^{-1}(t)g'(t)$.

Since $A^{-1}(t)g(t) = B'(t)$, we have $B''(t) = A^{-1}(t)[g'(t) - A'(t)B'(t)]$. And, $g'(t) = b''(t) - A''(t)B(t) - A'(t)B'(t)$, where $b''(t) = \frac{2D^T f}{n}$ and $A''(t) = \frac{2D^T D}{n}$.

Since $\|x_i\|_2, \|d_i\|_2, \|y_i\|_2, \|f_i\|_2 \leq c$, by Lemma A.1.10, there exists a constant $c_1$ such that $\|b'(t)\|_{op} \leq c_1$, $\|A'(t)\|_{op} \leq c_1$, $\|A''(t)\|_{op} \leq c_1$ and $\|b''(t)\|_{op} \leq c_1$, for all $t \in [0, k]$. Combined with the fact that $\|B(t)\|_{op} \lesssim \frac{1}{\lambda}$, we have $\|B''(t)\|_{op} \lesssim \frac{1}{\lambda} + \frac{1}{\lambda^2} + \frac{1}{\lambda^3}$ for $t \in [0, k]$.

### A.1.12 LEMMA A.1.12

Let $X, E \in \mathbb{R}^{n \times p}$ and $Y, F \in \mathbb{R}^{n \times q}$ be four matrices, and $\lambda > 0$ be a positive constant. Define $B_1 = \left( \frac{X^T X}{n} + \lambda I \right)^{-1} \frac{X^T Y}{n}$ and $B_2 = \left( \frac{(X+E)^T(X+E)}{n} + \lambda I \right)^{-1} \frac{(X+E)^T(Y+F)}{n}$, $\widehat{\Sigma}_n = \frac{X^T X}{n}$. Fix a positive constant $k > 0$, for any $\epsilon \in (0, k)$, define $D = E/\epsilon, f = F/\epsilon$, we have

$$\|B_1 - B_2\| \leq \frac{\epsilon}{\sigma_{min}(\widehat{\Sigma}_n) + \lambda} \left( \|\frac{X^T f}{n}\| + \|B_1\|_{op}\|\frac{X^T D}{n}\| + \|\frac{D^T(Y - XB_1)}{n}\| \right)$$

$$+ \left( \frac{1}{\lambda} + \frac{1}{\lambda^2} + \frac{1}{\lambda^3} \right) O(\epsilon^2).$$

where $\|\cdot\|$ is either $\|\cdot\|_{op}$ or $\|\cdot\|_F$.

**Proof:**

Consider the function $B(t) : \mathbb{R} \to \mathbb{R}^{p \times q}$ :

$$B(t) = \left\{ \frac{(X + tD)^T (X + tD)}{n} + \lambda I \right\}^{-1} \frac{1}{n} (X + tD)^T (Y + tf). \tag{20}$$

Since $\frac{(X+tD)^T(X+tD)}{n} + \lambda I$ is invertible for any $t$, $B(t)$ is infinitely differentiable, and by Taylor's theorem,

$$B(\epsilon) = B(0) + \epsilon \, B'(0) + O(\epsilon^2)$$

By Lemma A.1.11,

$$\|B(\epsilon) - B(0)\| \le \epsilon \|B'(0)\| + O\left( \left( \frac{1}{\lambda} + \frac{1}{\lambda^2} + \frac{1}{\lambda^3} \right) \epsilon^2 \right)$$

And,

$$\|B'(0)\| \le \| \left( \frac{X^T X}{n} + \lambda I \right)^{-1} \frac{X^T f}{n} \| + \| \left( \frac{X^T X}{n} + \lambda I \right)^{-1} \frac{X^T D \, B(0)}{n} \|$$

$$+ \| \left( \frac{X^T X}{n} + \lambda I \right)^{-1} \frac{D^T (Y - X \, B(0))}{n} \|$$

$$\le \frac{1}{\sigma_{min}(\widehat{\Sigma}_n) + \lambda} \left( \|\frac{X^T f}{n}\| + \|B(0)\|_{op} \|\frac{X^T D}{n}\| + \|\frac{D^T (Y - X \, B(0))}{n}\| \right)$$

Combining the above, for any $\epsilon \in (0, k)$,

$$\|B_1 - B_2\| = \|B(\epsilon) - B(0)\|$$

$$\le \frac{\epsilon}{\sigma_{min}(\widehat{\Sigma}_n) + \lambda} \left( \|\frac{X^T f}{n}\| + \|B_1\|_{op} \|\frac{X^T D}{n}\| + \|\frac{D^T (Y - X B_1)}{n}\| \right)$$

$$+ O\left( \left( \frac{1}{\lambda} + \frac{1}{\lambda^2} + \frac{1}{\lambda^3} \right) \epsilon^2 \right).$$

### A.1.13 LEMMA A.1.13

Let $Y \in \mathbb{R}^{n \times q}$ be a response matrix whose rows are transpose of $y_i$, $A \in \mathbb{R}^{n \times l_2}$ be a matrix whose rows are transpose of $v_i = a((W_1 + \Delta_1)\phi(x_i))$, $\widehat{A} \in \mathbb{R}^{n \times l_2}$ be a matrix whose rows are transpose of $\widehat{v}_i = a((W_1 + \widehat{\Delta}_1)\phi(x_i))$, and $E \in \mathbb{R}^{n,q}$ be a matrix whose rows are transpose of $\epsilon_i$. Denote the perturbed sample covariance, true sample covariance and the population covariance as $\widehat{\Sigma}_n = \frac{1}{n} \widehat{A}^T \widehat{A}$, $\Sigma_n = \frac{1}{n} A^T A$, and $\Sigma = \mathbb{E}[v_i v_i^T]$, respectively. We denote the ridge estimator using certain design matrix $\widetilde{A}$, the true and estimated projection matrices as

$$\widehat{B}(\widetilde{A}) := (\widetilde{A}^T \widetilde{A}/n + \lambda_2 I)^{-1} \frac{1}{n} \widetilde{A}^T (Y - \widetilde{A} W_2^T)$$

$$P_{k_2} := \mathcal{P}_{k_2}^r \left( \Delta_2 \Sigma (\Sigma + \lambda_2 I)^{-1} \Sigma \Delta_2^T \right)$$

$$\widehat{P}_{k_2} := \mathcal{P}_{k_2}^r \left( (Y - \widehat{A} W_2^T)^T \widehat{A} \widehat{B}(\widehat{A}) \right)$$

Define the terms

$$\Delta_v := v - \widehat{v} \qquad\qquad \Delta_A := A - \widehat{A}$$

$$\Delta_B := \widehat{B}(A) - \widehat{B}(\widehat{A}) \qquad\qquad \Delta_P := P_{k_2} - \widehat{P}_{k_2}$$

Then, the terms satisfy

$$\mathbb{E}\|\Delta_B\|_{op} \lesssim C_{\lambda_2} \mathbb{E}\|\Delta_1 - \widehat{\Delta}_1\|_{op},$$

$$\mathbb{E}\|\Delta_B\|_{op}^2 \lesssim C_{\lambda_2}^2 \mathbb{E}\|\Delta_1 - \widehat{\Delta}_1\|_{op}^2,$$

$$\mathbb{E}\|\Delta_P\|_{op} \lesssim \sqrt{k_2} K_{\lambda_2} \frac{1}{\sqrt{n}} + \sqrt{k_2} K_{\lambda_2} \mathbb{E}\|\Delta_1 - \widehat{\Delta}_1\|_{op},$$

$$\mathbb{E}\|\Delta_P\|_{op}^2 \lesssim k_2 K_{\lambda_2}^2 \frac{1}{n} + k_2 K_{\lambda_2}^2 \mathbb{E}\|\Delta_1 - \widehat{\Delta}_1\|_{op}^2 + k_2 K_{\lambda_2}^2 \sqrt{\frac{1}{n} \mathbb{E}\|\Delta_1 - \widehat{\Delta}_1\|_{op}^2},$$

where $C_{\lambda_2} = \left( \frac{1}{\lambda_2} + \frac{1}{\lambda_2^2} + \frac{1}{\lambda_2^3} \right), K_{\lambda_2} = \|\Delta_2\|_{op} (1 + \|\Delta_2\|_{op}) \left( 1 + \frac{1}{\lambda_2} \right)$.

**Proof:** First note that $\|\widehat{\Delta}_1\|_{op} \leq \|W_1\|_{op}$ is bounded. By assumption 4.2, $\|z_i\|_2 \leq c_1$, $\|\epsilon_i\|_2 \leq c_2$. By assumption 4.3, the activation function $a$ is $K$- Lipschitz continuous. Let $c_4$ be a constant such that $\|v\|_2 \leq K\|W_1 + \Delta_1\|_{op}\|z\|_2 \leq c_4\|W_1\|_{op}$ and $\|\widehat{v}\|_2 \leq K\|W_1 + \widehat{\Delta}_1\|_{op}\|z\|_2 \leq c_4\|W_1\|_{op}$.

By Lemma A.1.10, $\|\widehat{B}(A)\|_{op} \lesssim \frac{1}{\lambda_2}$. Define $\epsilon = \|\Delta_1 - \widehat{\Delta}_1\|_{op} \leq 2\|\Delta_1\|_{op} = k$, $D = -\Delta_A/\epsilon$, and $f = \Delta_A W_2^T/\epsilon$. The 2-norm of $D$'s rows are bounded by $\|v - \widehat{v}\|_2/\epsilon \leq Kc_1$ and that of $f$ are bounded by $K\|W_2\|_{op}c_1$. Apply Lemma A.1.12 with $D = -\Delta_A/\epsilon$, $f = \Delta_A W_2^T/\epsilon$, and $\epsilon = \|\Delta_1 - \widehat{\Delta}_1\|_{op}$, we have

$$\|\Delta_B\|_{op} \leq \frac{\|\Delta_1 - \widehat{\Delta}_1\|_{op}}{\lambda_2} \left( \|\frac{A^T f}{n}\|_{op} + 2\|\widehat{B}(A)\|_{op}\|\frac{A^T D}{n}\|_{op} + \|\frac{D^T (Y - AW_2^T)}{n}\|_{op} \right)$$

$$+ O\left( \left( \frac{1}{\lambda} + \frac{1}{\lambda^2} + \frac{1}{\lambda^3} \right) \|\Delta_1 - \widehat{\Delta}_1\|_{op}^2 \right),$$

for all $\epsilon \in (0, k)$. Since norm of each rows of $D$ and $f$ are bounded by a constant independent of $\epsilon$, by Lemma A.1.10, there exists a constant $c_6$ such that $\|\frac{A^T f}{n}\|, \|\frac{A^T D}{n}\|, \|\frac{D^T (Y - AW_2^T)}{n}\|, \|\frac{D^T A}{n}\| \leq c_6$. Combine with the fact that $\|\Delta_1 - \widehat{\Delta}_1\|_{op} \leq \|\Delta_1\|_{op} + \|W_1\|_{op}$, we have

$$\|\Delta_B\|_{op} \lesssim \|\Delta_1 - \widehat{\Delta}_1\|_{op} \left( \frac{1}{\lambda_2} + \frac{1}{\lambda_2^2} + \frac{1}{\lambda_2^3} \right)$$

$$\|\Delta_B\|_{op}^2 \lesssim \|\Delta_1 - \widehat{\Delta}_1\|_{op}^2 \left( \frac{1}{\lambda_2} + \frac{1}{\lambda_2^2} + \frac{1}{\lambda_2^3} \right)^2$$

$$\mathbb{E}\|\Delta_B\|_{op} \lesssim \mathbb{E}\|\Delta_1 - \widehat{\Delta}_1\|_{op} \left( \frac{1}{\lambda_2} + \frac{1}{\lambda_2^2} + \frac{1}{\lambda_2^3} \right)$$

$$\mathbb{E}\|\Delta_B\|_{op}^2 \lesssim \mathbb{E}\|\Delta_1 - \widehat{\Delta}_1\|_{op}^2 \left( \frac{1}{\lambda_2} + \frac{1}{\lambda_2^2} + \frac{1}{\lambda_2^3} \right)^2$$

For the term $\mathbb{E}\|\Delta_P\|_{op}$, let $S = \Delta_2\Sigma(\Sigma + \lambda_2 I)^{-1}\Sigma\Delta_2^T$ and $\widehat{S} = \frac{1}{n}(Y - \widehat{A}W_2^T)^T\widehat{A}\widehat{B}(\widehat{A})$. And, let $V$ be matrix whose columns are the top $k_2$ singular vectors of $S$, and $\widehat{V}$ be that of $\widehat{S}$. We have $P_{k_2} = VV^T$ and $\widehat{P}_{k_2} = \widehat{V}\widehat{V}^T$. By Lemma A.1.8, for any orthogonal matrix $R$,

$$\|\Delta_P\|_{op} = \|P_k - \widehat{P}_k\|_{op}$$
$$\leq 4\|\widehat{V}R - V\|_{op}$$

By Lemma A.1.6, there exists an orthogonal matrix $R$ s.t.,

$$\|\widehat{V}R - V\|_{op} \leq \frac{2^{\frac{3}{2}}\sqrt{k_2}\|S - \widehat{S}\|_{op}}{\lambda_{k_2}(S) - \lambda_{k_2+1}(S)}$$

Let $\widetilde{E} = E + \Delta_A(W_2 + \Delta_2)^T$, then $Y - \widehat{A}W_2^T = \widehat{A}\Delta_2^T + \widetilde{E}$, and

$$\widehat{S} = \frac{1}{n}(\widehat{A}\Delta_2^T + \widetilde{E})^T\widehat{A}(\widehat{\Sigma}_n + \lambda_2 I)^{-1}\frac{1}{n}\widehat{A}^T(\widehat{A}\Delta_2^T + \widetilde{E})$$

$$= \left( \Delta_2\widehat{\Sigma}_n + \frac{\widetilde{E}^T\widehat{A}}{n} \right)(\widehat{\Sigma}_n + \lambda_2 I)^{-1}\left( \widehat{\Sigma}_n\Delta_2^T + \frac{\widehat{A}^T\widetilde{E}}{n} \right)$$

We decompose the difference $\|S - \widehat{S}\|_{op}$ as follow

$$\|S - \widehat{S}\|_{op} \leq \|\Delta_2 \left( \widehat{\Sigma}_n(\widehat{\Sigma}_n + \lambda_2 I)^{-1}\widehat{\Sigma}_n - \Sigma(\Sigma + \lambda_2 I)^{-1}\Sigma \right) \Delta_2^T\|_{op}$$

$$+ 2\|\frac{\widetilde{E}^T\widehat{A}}{n}(\widehat{\Sigma}_n + \lambda_2 I)^{-1}\widehat{\Sigma}_n\Delta_2^T\|_{op}$$

$$+ \|\frac{\widetilde{E}^T\widehat{A}}{n}(\widehat{\Sigma}_n + \lambda_2 I)^{-1}\frac{\widehat{A}^T\widetilde{E}}{n}\|_{op}$$

By Lemma A.1.10 , there exists a constant $c_7$ such that $\|\frac{\widehat{A}^T \widetilde{E}}{n}\|_{op} \leq c_7$, $\|\frac{(W_2+\Delta_2)\Delta_A^T \widehat{A}}{n}\|_{op} \leq c_7\|\widehat{\Delta}_1 - \Delta_1\|_{op}$, $\|\frac{E^T \Delta_A}{n}\|_{op} \leq c_7\|\widehat{\Delta}_1 - \Delta_1\|_{op}$.

By inequalities $\|\widehat{\Sigma}_n\|_{op} \leq c_4^2$, $\|(\widehat{\Sigma}_n + \lambda_2 I)^{-1}\widehat{\Sigma}_n\|_{op} \leq \frac{c_4^2}{c_4^2+\lambda_2} \leq 1$, and $\|(\widehat{\Sigma}_n + \lambda_2 I)^{-1}\|_{op} \leq \frac{1}{\lambda_2}$, we have

$$\|\frac{\widetilde{E}^T \widehat{A}}{n}(\widehat{\Sigma}_n + \lambda_2 I)^{-1}\widehat{\Sigma}_n \Delta_2^T\|_{op} \leq \|\Delta_2\|_{op}\|\frac{\widetilde{E}^T \widehat{A}}{n}\|_{op}$$

$$\|\frac{\widetilde{E}^T \widehat{A}}{n}(\widehat{\Sigma}_n + \lambda_2 I)^{-1}\frac{\widehat{A}^T \widetilde{E}}{n}\|_{op} \leq \frac{c_7}{\lambda_2}\|\frac{\widetilde{E}^T \widehat{A}}{n}\|_{op}$$

And,

$$\|\frac{\widetilde{E}^T \widehat{A}}{n}\|_{op} = \|\frac{E^T \widehat{A}}{n}\|_{op} + \|\frac{(W_2 + \Delta_2)\Delta_A^T \widehat{A}}{n}\|_{op}$$

$$\leq \|\frac{E^T A}{n}\|_{op} + \|\frac{E^T \Delta_A}{n}\|_{op} + \|\frac{(W_2 + \Delta_2)\Delta_A^T \widehat{A}}{n}\|_{op}$$

$$\leq \|\frac{E^T A}{n}\|_{op} + 2c_7\|\widehat{\Delta}_1 - \Delta_1\|_{op}$$

Since $\epsilon_i$ and $v_i$ are independent, by (3) of Lemma A.1.10, $\mathbb{E}\|\frac{E^T A}{n}\|_{op} \lesssim \frac{1}{\sqrt{n}}$, $\mathbb{E}\|\frac{E^T A}{n}\|_{op}^2 \lesssim \frac{1}{n}$. Hence,

$$\|S - \widehat{S}\|_{op}$$

$$\lesssim \|\Delta_2 \left(\widehat{\Sigma}_n(\widehat{\Sigma}_n + \lambda_2 I)^{-1}\widehat{\Sigma}_n - \Sigma(\Sigma + \lambda_2 I)^{-1}\Sigma\right)\Delta_2^T\|_{op}$$

$$+ \|\Delta_2\|_{op}\left(1 + \frac{1}{\lambda_2}\right)\|\frac{\widetilde{E}^T \widehat{A}}{n}\|_{op}$$

To bound the first term, let $K_n = \widehat{\Sigma}_n(\widehat{\Sigma}_n + \lambda_2 I)^{-1}$ and $K = \Sigma(\Sigma + \lambda_2 I)^{-1}$, then

$$\|K_n - K\|_{op} = \|\widehat{\Sigma}_n(\widehat{\Sigma}_n + \lambda_2 I)^{-1} - \Sigma(\Sigma + \lambda_2 I)^{-1}\|_{op}$$

$$\leq \|\widehat{\Sigma}_n(\widehat{\Sigma}_n + \lambda_2 I)^{-1} - \widehat{\Sigma}_n(\Sigma + \lambda_2 I)^{-1}\|_{op}$$

$$+ \|\widehat{\Sigma}_n(\Sigma + \lambda_2 I)^{-1} - \Sigma(\Sigma + \lambda_2 I)^{-1}\|_{op}$$

$$\|\widehat{\Sigma}_n(\widehat{\Sigma}_n + \lambda_2 I)^{-1} - \widehat{\Sigma}_n(\Sigma + \lambda_2 I)^{-1}\|_{op} = \|\widehat{\Sigma}_n\left((\widehat{\Sigma}_n + \lambda_2 I)^{-1}(\Sigma - \widehat{\Sigma}_n)(\Sigma + \lambda_2 I)^{-1}\right)\|_{op}$$

$$\leq \frac{1}{\lambda_2}\|\Sigma - \widehat{\Sigma}_n\|_{op}$$

$$\|\widehat{\Sigma}_n(\Sigma + \lambda_2 I)^{-1} - \Sigma(\Sigma + \lambda_2 I)^{-1}\|_{op} = \|(\widehat{\Sigma}_n - \Sigma)(\Sigma + \lambda_2 I)^{-1}\|_{op}$$

$$\leq \frac{1}{\lambda_2}\|\Sigma - \widehat{\Sigma}_n\|_{op}$$

Hence, $\|K_n - K\|_{op} \lesssim \frac{1}{\lambda_2}\|\Sigma - \widehat{\Sigma}_n\|_{op}$. Using $\|K_n\|_{op} \leq 1$, $\|K\|_{op} \leq 1$, $\|\widehat{\Sigma}_n\|_{op} \leq c_4^2$, we bound the first term by

$$\|\Delta_2\left(\widehat{\Sigma}_n(\widehat{\Sigma}_n + \lambda_2 I)^{-1}\widehat{\Sigma}_n - \Sigma(\Sigma + \lambda_2 I)^{-1}\Sigma\right)\Delta_2^T\|_{op}$$

$$\leq \|\Delta_2\|_{op}^2\|\widehat{\Sigma}_n(\widehat{\Sigma}_n + \lambda_2 I)^{-1}\widehat{\Sigma}_n - \Sigma(\Sigma + \lambda_2 I)^{-1}\Sigma\|_{op}$$

$$\leq \|\Delta_2\|_{op}^2\|K_n\widehat{\Sigma}_n - K\Sigma\|_{op}$$

$$= \|\Delta_2\|_{op}^2\|(K_n - K)\widehat{\Sigma}_n + K(\widehat{\Sigma}_n - \Sigma)\|_{op}$$

$$\lesssim \|\Delta_2\|_{op}^2 \frac{1}{\lambda_2}\|\widehat{\Sigma}_n - \Sigma\|_{op} + \|\Delta_2\|_{op}^2\|\widehat{\Sigma}_n - \Sigma\|_{op}$$

$$\lesssim \|\Delta_2\|_{op}^2\left(1 + \frac{1}{\lambda_2}\right)\|\widehat{\Sigma}_n - \Sigma\|_{op}$$

To bound the difference in covariance, we decompose as follow

$$\|\widehat{\Sigma}_n - \Sigma\|_{op} \leq \|\Sigma_n - \Sigma\|_{op} + 2\|\frac{\Delta_A^T A}{n}\|_{op} + \|\frac{\Delta_A^T \Delta_A}{n}\|_{op}$$

$$\lesssim \|\Sigma_n - \Sigma\|_{op} + \|\Delta_1 - \widehat{\Delta}_1\|_{op} + \|\Delta_1 - \widehat{\Delta}_1\|_{op}^2$$

$$\lesssim \|\Sigma_n - \Sigma\|_{op} + \|\Delta_1 - \widehat{\Delta}_1\|_{op}$$

Since $\|v_i\|_2^2$ are bounded, by Theorem 4.7.1 in Vershynin (2018), $\mathbb{E}\|\Sigma_n - \Sigma\|_{op} = O\left(\frac{1}{\sqrt{n}}\right)$, thus

$$\mathbb{E}\|\widehat{\Sigma}_n - \Sigma\|_{op} \lesssim \frac{1}{\sqrt{n}} + \mathbb{E}\|\Delta_1 - \widehat{\Delta}_1\|_{op}$$

Let $K_{\lambda_2} = \left(1 + \frac{1}{\lambda_2}\right)$ and apply Cauchy-Schwarz $\mathbb{E}[ab] \leq \sqrt{\mathbb{E}[a^2]\mathbb{E}[b^2]}$ for $\mathbb{E}\|S - \widehat{S}\|_{op}^2$,

$$\mathbb{E}\|S - \widehat{S}\|_{op} \lesssim K_{\lambda_2}\frac{1}{\sqrt{n}} + K_{\lambda_2}\mathbb{E}\|\Delta_1 - \widehat{\Delta}_1\|_{op}$$

$$\mathbb{E}\|S - \widehat{S}\|_{op}^2 \lesssim K_{\lambda_2}^2\frac{1}{n} + K_{\lambda_2}^2\mathbb{E}\|\Delta_1 - \widehat{\Delta}_1\|_{op}^2 + K_{\lambda_2}^2\sqrt{\frac{1}{n}\mathbb{E}\|\Delta_1 - \widehat{\Delta}_1\|_{op}^2}$$

Thus,

$$\mathbb{E}\|\Delta_P\|_{op} \lesssim \sqrt{k_2}\mathbb{E}\|S - \widehat{S}\|_{op}$$

$$\lesssim \sqrt{k_2}K_{\lambda_2}\frac{1}{\sqrt{n}} + \sqrt{k_2}K_{\lambda_2}\mathbb{E}\|\Delta_1 - \widehat{\Delta}_1\|_{op}$$

$$\mathbb{E}\|\Delta_P\|_{op}^2 \lesssim k_2 K_{\lambda_2}^2\frac{1}{n} + k_2 K_{\lambda_2}^2\mathbb{E}\|\Delta_1 - \widehat{\Delta}_1\|_{op}^2 + k_2 K_{\lambda_2}^2\sqrt{\frac{1}{n}\mathbb{E}\|\Delta_1 - \widehat{\Delta}_1\|_{op}^2}$$

### A.1.14 THEOREM 4.1

We reuse the definitions of key quantities in A.1.13, and additionally define $\widetilde{y} := \Delta_2 v + \epsilon$ and $\widetilde{Y} := A\Delta_2^T + E$. The prediction given $x$ is $\widehat{y} := (W_2 + \widehat{\Delta}_2)\widehat{v}$. We decompose the prediction error as follow,

$$\|y - (W_2 + \widehat{\Delta}_2)\widehat{v}\|_2^2 \lesssim \|y - (W_2 + \widehat{\Delta}_2)v\|_2^2 + \|(W_2 + \widehat{\Delta}_2)\Delta_v\|_2^2$$

$$= \|\widetilde{y} - \widehat{\Delta}_2 v\|_2^2 + \|W_2 + \widehat{\Delta}_2\|_{op}^2\|\Delta_v\|_2^2$$

$$\|\widetilde{y} - \widehat{\Delta}_2 v\|_2^2 = \|\widetilde{y} - \widehat{P}_{k_2}\widehat{B}(\widehat{A})^T v\|_2^2$$

$$\lesssim \|\widetilde{y} - \widehat{P}_{k_2}\widehat{B}(A)^T v\|_2^2 + \|\widehat{P}_{k_2}\Delta_B^T v\|_2^2$$

$$\lesssim \|\widetilde{y} - \widehat{P}_{k_2}\widehat{B}(A)^T v\|_2^2 + \|\Delta_B\|_{op}^2\|v\|_2^2$$

$$\|\widetilde{y} - \widehat{P}_{k_2}\widehat{B}(A)^T v\|_2^2 \lesssim \|\widetilde{y} - P_{k_2}\widehat{B}(A)^T v\|_2^2 + \|\Delta_P\widehat{B}(A)^T v\|_2^2$$

$$\leq \|\widetilde{y} - P_{k_2}\widehat{B}(A)^T v\|_2^2 + \|\Delta_P\|_{op}^2\|\widehat{B}(A)\|_{op}^2\|v\|_2^2$$

$$\|\widetilde{y} - P_{k_2}\widehat{B}(A)^T v\|_2^2 = \|P_{k_2}\widetilde{y} + (I - P_{k_2})\widetilde{y} - P_{k_2}\widehat{B}(A)^T v\|_2^2$$

$$= \|P_{k_2}\widetilde{y} - P_{k_2}\widehat{B}(A)^T v\|_2^2 + \|(I - P_k)\widetilde{y}\|_2^2$$

By assumption 4.2 and 4.3, $\|z_i\|_2 \leq c_1$, $\|\epsilon_i\|_2 \leq c_2$. We let $c_4$ be a constant such that $\|v\|_2 \leq K\|W_1 + \Delta_1\|_{op}\|z\|_2 \leq c_4$ and $\|\widehat{v}\|_2 \leq K\|W_1 + \widehat{\Delta}_1\|_{op}\|z\|_2 \leq c_4$. By Lemma A.1.10, $\|\widehat{B}(A)\|_{op} \lesssim \frac{1}{\lambda_2}$. Therefore,

$$\mathbb{E}\|y - (W_2 + \widehat{\Delta}_2)\widehat{v}\|_2^2 \lesssim \left(1 + \frac{1}{\lambda_2}\right)^2\mathbb{E}\|\Delta_v\|_2^2 + \mathbb{E}\|\Delta_B\|_{op}^2 + \left(\frac{1}{\lambda_2}\right)^2\mathbb{E}\|\Delta_P\|_{op}^2$$

$$+ \mathbb{E}\|P_{k_2}\widetilde{y} - P_{k_2}\widehat{B}(A)^T v\|_2^2 + \mathbb{E}\|(I - P_{k_2})\widetilde{y}\|_2^2$$

By the Lipschitz-continuity of activation function $a$,

$$\mathbb{E}\|\Delta_v\|_2^2 \leq K^2 \mathbb{E}\|(\Delta_1 - \widehat{\Delta}_1)z\|_2^2$$
$$\lesssim \mathbb{E}\|\Delta_1 - \widehat{\Delta}_1\|_{op}^2$$

By Lemma A.1.13,

$$\mathbb{E}\|\Delta_B\|_{op}^2 \lesssim C_{\lambda_2}^2 \mathbb{E}\|\Delta_1 - \widehat{\Delta}_1\|_{op}^2$$

$$\mathbb{E}\|\Delta_P\|_{op}^2 \lesssim k_2 K_{\lambda_2}^2 \frac{1}{n} + k_2 K_{\lambda_2}^2 \mathbb{E}\|\Delta_1 - \widehat{\Delta}_1\|_{op}^2 + k_2 K_{\lambda_2}^2 \sqrt{\frac{1}{n} \mathbb{E}\|\Delta_1 - \widehat{\Delta}_1\|_{op}^2}$$

where $C_{\lambda_2} = \left(\frac{1}{\lambda_2} + \frac{1}{\lambda_2^2} + \frac{1}{\lambda_2^3}\right), K_{\lambda_2} = \left(1 + \frac{1}{\lambda_2}\right)$.

For the bias term,

$$\mathbb{E}\|(I - P_{k_2})\widetilde{y}\|_2^2 = \mathbb{E}\|(I - P_{k_2})\Delta_2 v + (I - P_{k_2})\epsilon\|_2^2$$
$$= \mathbb{E}\|(I - P_{k_2})\Delta_2 v\|_2^2 + \mathbb{E}\|(I - P_{k_2})\epsilon\|_2^2$$
$$\mathbb{E}\|(I - P_{k_2})\Delta_2 v\|_2^2 = \mathbb{E}\mathrm{Tr}((I - P_{k_2})\Delta_2 v v^T \Delta_2^T (I - P_{k_2}))$$
$$= \mathrm{Tr}((I - P_{k_2})\Delta_2 \Sigma \Delta_2^T (I - P_{k_2}))$$
$$= \|(I - P_{k_2})\Delta_2 \Sigma^{\frac{1}{2}}\|_F^2$$

Finally, consider the term

$$\mathbb{E}\|P_{k_2}\widetilde{y} - P_{k_2}\widehat{B}(A)^T v\|_2^2 = \mathbb{E}\|P_{k_2}\widetilde{y} - P_{k_2}\widetilde{Y}^T \frac{1}{n} A(\Sigma_n + \lambda_2 I)^{-1}\|_2^2$$

This is the random design prediction error of ridge regression estimator with response $P_{k_2}\widetilde{y}$, design matrix $A$, and noise term $P_{k_2}\epsilon$. By Theorem 1 in Mourtada & Rosasco (2022),

$$\mathbb{E}\|P_{k_2}\widetilde{y} - P_{k_2}\widehat{B}(Z)^T z\|_2^2 \leq \lambda_2 \left(1 + \frac{c_1^2}{\lambda_2 n}\right)^2 \|(\Sigma + \lambda_2)^{-\frac{1}{2}} \Sigma^{\frac{1}{2}} P_{k_2} \Delta_2\|_F^2$$
$$+ \left(1 + \frac{c_1^2}{\lambda_2 n}\right) \frac{\mathrm{Tr}((\Sigma + \lambda_2)^{-1}\Sigma)}{n} \mathbb{E}\|P_{k_2}\epsilon\|_2^2$$
$$+ \mathbb{E}\|P_{k_2}\epsilon\|_2^2$$

where $\mathbb{E}\|P_{k_2}\epsilon\|_2^2 = \mathrm{Tr}(\mathbb{E}[\epsilon\epsilon^T]P_{k_2})$. And, $\mathbb{E}\|(I - P_{k_2})\epsilon\|_2^2 + \mathbb{E}\|P_{k_2}\epsilon\|_2^2 = \mathbb{E}\|\epsilon\|_2^2$. Combining the above,

$$L(\widehat{\Delta}_1, \widehat{\Delta}_2) = \mathbb{E}\|y - (W_2 + \widehat{\Delta}_2)\widehat{z}\|_2^2$$
$$\lesssim \left(\left(1 + \frac{1}{\lambda_2}\right)^2 + C_{\lambda_2}^2 + k_2 \left(\frac{K_{\lambda_2}}{\lambda_2}\right)^2\right) \mathbb{E}\|\Delta_1 - \widehat{\Delta}_1\|_{op}^2$$
$$+ k_2 \left(\frac{K_{\lambda_2}}{\lambda_2}\right)^2 \frac{1}{n}$$
$$+ k_2 \left(\frac{K_{\lambda_2}}{\lambda_2}\right)^2 \left(\sqrt{\frac{1}{n} \mathbb{E}\|\Delta_1 - \widehat{\Delta}_1\|_{op}^2}\right)$$
$$+ \lambda_2 \left(1 + \frac{1}{\lambda_2 n}\right)^2 \|(\Sigma + \lambda_2)^{-\frac{1}{2}} \Sigma^{\frac{1}{2}} P_{k_2} \Delta_2\|_F^2$$
$$+ \left(1 + \frac{1}{\lambda_2 n}\right) \frac{\mathrm{Tr}((\Sigma + \lambda_2)^{-1}\Sigma)}{n} \mathrm{Tr}(\mathbb{E}[\epsilon\epsilon^T]P_{k_2})$$
$$+ \|(I - P_{k_2})\Delta_2 \Sigma^{\frac{1}{2}}\|_F^2$$
$$+ \mathbb{E}\|\epsilon\|_2^2$$

Finally, the excess risk is

$$\mathcal{E}(\widehat{\Delta}_1, \widehat{\Delta}_2) = L(\widehat{\Delta}_1, \widehat{\Delta}_2) - L(\Delta_1, \Delta_2)$$
$$= L(\widehat{\Delta}_1, \widehat{\Delta}_2) - \mathbb{E}\|\epsilon\|_2^2$$
$$= L(\widehat{\Delta}_1, \widehat{\Delta}_2) - \mathrm{Tr}(\mathbb{E}[\epsilon\epsilon^T]).$$

We define

$$\Delta_{\text{stat}} = k_2 \left(\frac{K_{\lambda_2}}{\lambda_2}\right)^2 \left(\frac{1}{n} + \sqrt{\frac{1}{n}\mathbb{E}\|\Delta_1 - \widehat{\Delta}_1\|_{op}^2}\right) + \left(1 + \frac{1}{\lambda_2 n}\right)\frac{\mathrm{Tr}((\Sigma + \lambda_2)^{-1}\Sigma)}{n}\mathrm{Tr}(\mathbb{E}[\epsilon\epsilon^T]P_{k_2}),$$

$$\Delta_{\text{first}} = \left(\left(1 + \frac{1}{\lambda_2}\right)^2 + C_{\lambda_2}^2 + k_2\left(\frac{K_{\lambda_2}}{\lambda_2}\right)^2\right)\mathbb{E}\|\Delta_1 - \widehat{\Delta}_1\|_{op}^2,$$

$$\Delta_{\text{bias}} = \lambda_2\left(1 + \frac{1}{\lambda_2 n}\right)^2\|(\Sigma + \lambda_2)^{-\frac{1}{2}}\Sigma^{\frac{1}{2}}P_{k_2}\Delta_2\|_F^2 + \|(I - P_{k_2})\Delta_2\Sigma^{\frac{1}{2}}\|_F^2.$$

Then, $\mathcal{E}(\widehat{\Delta}_1, \widehat{\Delta}_2) \lesssim \Delta_{\text{stat}} + \Delta_{\text{first}} + \Delta_{\text{bias}}$

### A.1.15 THEOREM 4.2

Suppose model (1), assumption 4.1, and 4.2 holds. Then,

$$E\|\widehat{\Delta}_1 - \Delta_1\|_F \lesssim \|\widehat{\Delta}_1 - \Delta_1\|_F \lesssim \Delta_{\text{stat}} + \Delta_{bias}$$

where

$$\Delta_{\text{stat}} = \frac{1}{1 + \lambda_1}\frac{\sqrt{k_1}l_1}{\sqrt{n}}$$

$$\Delta_{\text{bias}} = \frac{1}{1 + \lambda_1}\delta + \frac{\lambda_1}{1 + \lambda_1}\|\Delta_1\|_F + \frac{1}{1 + \lambda_1}\|W_1(VV^T - I)(I - P_{k_1})\|_F$$

**Proof:**
Let

$$B = \left[\mathbb{E}[Y_{i1}S(Z_i)], \ldots, \mathbb{E}[Y_{iq}S(Z_i)]\right]$$

$$\widehat{B} = \frac{1}{n}\left[\sum_{i=1}^n Y_{i1}S(Z_i), \ldots, \sum_{i=1}^n Y_{iq}S(Z_i)\right]$$

$$V = \mathrm{SVD}_l(B)$$

$$\widehat{V} = \mathrm{SVD}_l\left(\widehat{B}\right)$$

$$V_* = \mathrm{SVD}_r(W_1 + \Delta_1)$$

By 4.2 and model (1), the conditions in Lemma 3.1 hold. By assumption 4.1 and Lemma 3.1, $\mathrm{Col}(B) = \mathrm{Row}(W_1 + \Delta_1)$. By the uniqueness of orthogonal projection matrices, $VV^T = V_*V_*^T$.

We first consider $\widehat{\Delta}_1^* = -W_1(I - \widehat{V}\widehat{V}^T)$, the first layer estimator with $\lambda_1 = 0$ and $k_1 \geq \min\{l_1, l_2\}$. From the equality $W_1 + \Delta_1 = W_1VV^T + \Delta_1VV^T$, we have $\Delta_1 = W_1(VV^T - I) + \Delta_1VV^T$. Recall that $\|\Delta_1VV^T\|_F = \delta$, so

$$\|\widehat{\Delta}_1^* - \Delta_1\|_F = \|\widehat{\Delta}_1^* - W_1(VV^T - I) - \Delta_1VV^T\|_F$$
$$\leq \|\widehat{\Delta}_1^* - W_1(VV^T - I)\|_F + \|\Delta_1VV^T\|_F$$
$$= \|\widehat{\Delta}_1^* - W_1(VV^T - I)\|_F + \delta$$

For the first term,

$$\|\widehat{\Delta}_1^* - W_1(VV^T - I)\|_F \leq \|W_1\|_{op}\|\widehat{V}\widehat{V}^T - VV^T\|_F$$

Let $E = B - \widehat{B}$, $r_V = \text{rank}(B) = \text{rank}(W_1 + \Delta_1)$. By Lemma A.1.5, there exists a constant $C > 0$ such that $\|E\|_{op} \leq C\sqrt{q}$ almost surely. By Lemma A.1.9,

$$\mathbb{E}\|\widehat{V}\widehat{V}^T - VV^T\|_F \lesssim \mathbb{E}\|E\|_{op}$$

And, for any $t \geq 0$, $\mathbb{P}\left(\|E\|_{op} \geq t\right) \leq 2l_1 \exp\left(\frac{-nt^2/2}{C^2+Ct/3}\right)$. When $t > 3C$, $C^2 < Ct/3$ and $\exp(\frac{-nt^2/2}{C^2+Ct/3}) \leq \exp(\frac{-nt^2/2}{2Ct/3})$. Otherwise, $\exp(\frac{-nt^2/2}{C^2+Ct/3}) \leq \exp(\frac{-nt^2/2}{2C^2})$. Using the integrated tail formula,

$$\begin{aligned}
\mathbb{E}\|E\|_{op} &= \int_0^\infty \mathbb{P}\left(\|E\|_{op} \geq t\right) dt \\
&\leq \int_0^{3C} 2l_1 \exp\left(-\frac{n}{4C^2}t^2\right) dt + \int_{3C}^\infty 2l_1 \exp\left(-\frac{3n}{4C}t\right) dt \\
&\leq \int_0^\infty 2l_1 \exp\left(-\frac{n}{4C^2}t^2\right) dt + \int_0^\infty 2l_1 \exp\left(-\frac{3n}{4C}t\right) dt \\
&\lesssim \frac{l_1}{\sqrt{n}} + \frac{l_1}{n}
\end{aligned}$$

And so $\mathbb{E}\|\widehat{\Delta}_1^* - \Delta_1\|_F \lesssim \frac{l_1}{\sqrt{n}} + \delta$. Now we consider the general estimator $\widehat{\Delta}_1 = \frac{1}{1+\lambda_1}\mathcal{R}_{k_1}\left(\widehat{\Delta}_1^*\right)$. Let $P_{k_1} = \mathcal{P}_{k_1}^r(-W_1(I - VV^T))$ and $\widehat{P}_{k_1} = \mathcal{P}_{k_1}^r(\widehat{\Delta}_1^*)$. Then,

$$\begin{aligned}
\|\widehat{\Delta}_1 - \Delta_1\|_F &= \|\frac{1}{1+\lambda_1}\widehat{\Delta}_1^*\widehat{P}_{k_1} - \Delta_1\|_F \\
&\leq \frac{1}{1+\lambda_1}\|\widehat{\Delta}_1^*\widehat{P}_{k_1} - \Delta_1\|_F + \frac{\lambda_1}{1+\lambda_1}\|\Delta_1\|_F \\
&\leq \frac{1}{1+\lambda_1}\|\widehat{\Delta}_1^*\widehat{P}_{k_1} - W_1(VV^T - I)\|_F + \frac{1}{1+\lambda_1}\|\Delta_1 VV^T\|_F + \frac{\lambda_1}{1+\lambda_1}\|\Delta_1\|_F
\end{aligned}$$

Further decompose the first term,

$$\|\widehat{\Delta}_1^*\widehat{P}_{k_1} - W_1(VV^T - I)\|_F \leq \|\widehat{\Delta}_1^*\widehat{P}_{k_1} - W_1(VV^T - I)P_{k_1}\|_F + \|W_1(VV^T - I)(I - P_{k_1})\|_F$$

$$\begin{aligned}
\mathbb{E}\|\widehat{\Delta}_1^*\widehat{P}_{k_1} - W_1(VV^T - I)P_{k_1}\|_F &\leq \mathbb{E}\|(\widehat{\Delta}_1^* - W_1(VV^T - I))P_{k_1}\|_F + \mathbb{E}\|\widehat{\Delta}_1^*(\widehat{P}_{k_1} - P_{k_1})\|_F \\
&\lesssim \mathbb{E}\|\widehat{\Delta}_1^* - W_1(VV^T - I)\|_F\|P_{k_1}\|_F + \mathbb{E}\|\widehat{\Delta}_1^*(\widehat{P}_{k_1} - P_{k_1})\|_F \\
&\lesssim \frac{l_1\sqrt{k_1}}{\sqrt{n}} + \|W_1\|_{op}\mathbb{E}\|\widehat{P}_{k_1} - P_{k_1}\|_F
\end{aligned}$$

Apply Lemma A.1.9 twice, $\mathbb{E}\|\widehat{P}_{k_1} - P_{k_1}\|_F \lesssim \sqrt{k_1}\mathbb{E}\|E\|_{op} \lesssim \frac{\sqrt{k_1}l_1}{\sqrt{n}}$. Combining all,

$$\mathbb{E}\|\widehat{\Delta}_1 - \Delta_1\|_F \lesssim \Delta_{\text{stat}} + \Delta_{bias}$$

where

$$\Delta_{\text{stat}} = \frac{1}{1+\lambda_1}\frac{\sqrt{k_1}}{\sqrt{n}}$$

$$\Delta_{\text{bias}} = \frac{1}{1+\lambda_1}\delta + \frac{\lambda_1}{1+\lambda_1}\|\Delta_1\|_F + \frac{1}{1+\lambda_1}\|W_1(VV^T - I)(I - P_{k_1})\|_F$$

Finally, by nature of the projection matrix $P_{k_1}$, $\|W_1(VV^T - I)(I - P_{k_1})\|_F = \sqrt{\sum_{i>k_1}\sigma_i^2(W_1(VV^T - I))}$.

### A.1.16 THEOREM 4.3

Assume the model (1) and assumption (4.2) hold. And $\widehat{\Delta}_2$ be in (11), then

$$\mathbb{E}\|\widehat{\Delta}_2 - \Delta_2\|_F \lesssim \Delta_{\text{bias}} + \Delta_{\text{stat}} + \Delta_{\text{first}}.$$

where

$$\Delta_{\text{bias}} = \|(I - P_k)\Delta_2\|_F + \|P_k\Delta_2\left(I - \Sigma_n(\Sigma_n + \lambda_2 I)^{-1}\right)\|_F$$

$$\Delta_{\text{stat}} = \left(\frac{K_{\lambda_2}}{\lambda_2}\sqrt{k_2} + \frac{1}{\lambda_2}\sqrt{\text{Tr}(\mathbb{E}[\epsilon_i\epsilon_i^T]P_k)}\right)\frac{1}{\sqrt{n}}$$

$$\Delta_{\text{first}} = \left(\frac{K_{\lambda_2}}{\lambda_2} + C_{\lambda_2}\right)\mathbb{E}\|\Delta_1 - \widehat{\Delta}_1\|_{op}$$

$C_{\lambda_2} = \left(\frac{1}{\lambda_2} + \frac{1}{\lambda_2^2} + \frac{1}{\lambda_2^3}\right)$ and $K_{\lambda_2} = \left(1 + \frac{1}{\lambda_2}\right)$.

**Proof:** We reuse the definitions of key quantities in A.1.13, and decompose the error as follow

$$\|\Delta_2 - \widehat{\Delta}_2\|_F = \|\Delta_2 - \widehat{P}_{k_2}\widehat{B}(\widehat{Z})^T\|_F$$
$$\leq \|(I - P_{k_2})\Delta_2\|_F + \|P_{k_2}\Delta_2 - \widehat{P}_{k_2}\widehat{B}(\widehat{Z})^T\|_F$$

By Lemma A.1.10, $\|\widehat{B}(A)\|_F \lesssim \frac{1}{\lambda_2}$. The remaining term can be bounded as

$$\|P_{k_2}\Delta_2 - \widehat{P}_{k_2}\widehat{B}(\widehat{A})^T\|_F \leq \|P_{k_2}\Delta_2 - \widehat{P}_{k_2}\widehat{B}(A)^T\|_F + \|\widehat{P}_{k_2}\Delta_B\|_F$$
$$\leq \|P_{k_2}\Delta_2 - P_{k_2}\widehat{B}(A)^T\|_F + \|\Delta_P\widehat{B}(A)\|_F + \|\widehat{P}_{k_2}\Delta_B\|_F$$
$$\leq \|P_{k_2}\Delta_2 - P_{k_2}\widehat{B}(A)^T\|_F + \|\Delta_P\|_{op}\|\widehat{B}(A)\|_F + \|\Delta_B\|_{op}\|\widehat{P}_{k_2}\|_F$$
$$\lesssim \|P_{k_2}\Delta_2 - P_{k_2}\widehat{B}(A)^T\|_F + \|\Delta_P\|_{op}\frac{1}{\lambda_2} + \sqrt{k_2}\|\Delta_B\|_{op}$$

Using $Y - AW_2^T = A\Delta_2^T + E$, the first term can be bounded as

$$\|P_{k_2}\Delta_2 - P_{k_2}\widehat{B}(A)^T\|_F = \|P_{k_2}\Delta_2 - P_{k_2}\left(\Delta_2 A^T + E^T\right)\frac{1}{n}A(\Sigma_n + \lambda_2 I)^{-1}\|_F$$
$$\leq \|P_{k_2}\Delta_2\left(I - \Sigma_n(\Sigma_n + \lambda_2 I)^{-1}\right)\|_F + \|\frac{P_{k_2}E^T A}{n}(\Sigma_n + \lambda_2 I)^{-1})\|_F$$

$$\|\frac{P_{k_2}E^T A}{n}(\Sigma_n + \lambda_2 I)^{-1}\|_F \leq \|(\Sigma_n + \lambda_2 I)^{-1}\|_{op}\|\frac{P_{k_2}E^T A}{n}\|_F$$
$$\leq \frac{1}{\sigma_{min}(\Sigma_n) + \lambda_2}\|\frac{P_{k_2}E^T A}{n}\|_F$$
$$\leq \frac{1}{\lambda_2}\|\frac{P_{k_2}E^T A}{n}\|_F$$

By (3) of Lemma A.1.10,

$$\mathbb{E}\|\frac{P_{k_2}E^T Z}{n}\|_F \leq \sqrt{\mathbb{E}\|\frac{P_{k_2}E^T Z}{n}\|_F^2}$$
$$\leq \frac{c_1\sqrt{\mathbb{E}\|P_{k_2}\epsilon_i\|_2^2}}{\sqrt{n}}$$
$$\sqrt{\mathbb{E}\|P_{k_2}\epsilon_i\|_2^2} = \sqrt{\text{Tr}(\mathbb{E}[\epsilon_i\epsilon_i^T]P_{k_2})}$$

By Lemma A.1.13,

$$\mathbb{E}\|\Delta_B\|_{op} \lesssim C_{\lambda_2}\mathbb{E}\|\Delta_1 - \widehat{\Delta}_1\|_{op}$$
$$\mathbb{E}\|\Delta_P\|_{op} \lesssim \sqrt{k_2}K_{\lambda_2}\frac{1}{\sqrt{n}} + \sqrt{k_2}K_{\lambda_2}\mathbb{E}\|\Delta_1 - \widehat{\Delta}_1\|_{op}$$

where $C_{\lambda_2} = \left(\frac{1}{\lambda_2} + \frac{1}{\lambda_2^2} + \frac{1}{\lambda_2^3}\right), K_{\lambda_2} = \left(1 + \frac{1}{\lambda_2}\right)$. Finally, we let

$$\Delta_{\text{bias}} = \|(I - P_k)\Delta_2\|_F + \mathbb{E}\|P_k\Delta_2\left(I - \Sigma_n(\Sigma_n + \lambda_2 I)^{-1}\right)\|_F$$

$$\Delta_{\text{stat}} = \left(\frac{K_{\lambda_2}}{\lambda_2}\sqrt{k_2} + \frac{1}{\lambda_2}\sqrt{\text{Tr}(\mathbb{E}[\epsilon_i \epsilon_i^T]P_k)}\right)\frac{1}{\sqrt{n}}$$

$$\Delta_{\text{first}} = \left(\frac{K_{\lambda_2}}{\lambda_2} + C_{\lambda_2}\right)\mathbb{E}\|\Delta_1 - \widehat{\Delta}_1\|_{op}$$

And,

$$\mathbb{E}\|\widehat{\Delta}_2 - \Delta_2\|_F \lesssim \Delta_{\text{bias}} + \Delta_{\text{stat}} + \Delta_{\text{first}}.$$

