# OpenReview forum: "Consistent Low-Rank Adaptation of Two-layer Neural Networks: A Nonparametric Statistics Approach"
_ICLR.cc/2026/Conference — ICLR 2026 Conference Withdrawn Submission_

### Official Review · Reviewer_2EjR · 2025-10-31

**Soundness:** 2
**Presentation:** 2
**Contribution:** 1
**Rating:** 2
**Confidence:** 3

**Summary:**

The paper investigates LoRA (Low-Rank Adaptation) in the context of a two-layer neural network. By applying Stein’s second-order lemma, the authors derive a closed-form solution, eliminating the need for iterative SGD optimization and offering significant computational efficiency. Experiments on MNIST are presented to demonstrate the effectiveness of the proposed solution.

**Strengths:**

*  The paper takes a clever and analytically tractable approach by simplifying the setup to make use of Stein’s second-order lemma.
*  The approach provides computational advantages, avoiding iterative training loops.

**Weaknesses:**

*  The formulation may be an over-simplification, limiting the practical applicability of the findings to real-world deep networks.
*  As a result, the theoretical contribution might have narrow scope, primarily useful as a pedagogical or illustrative case rather than as a general method.
*  Experimental validation is weak: MNIST is a toy dataset, and no additional empirical evidence is provided to support generalization beyond simple settings.
*  The claim that the closed-form solution can serve as a strong initialization for LoRA is not empirically verified.

**Questions:**

See weakness, and
1. Line 131: The function $f$ appears to be undefined.

---

### Official Review · Reviewer_tHrb · 2025-11-01

**Soundness:** 3
**Presentation:** 2
**Contribution:** 1
**Rating:** 2
**Confidence:** 3

**Summary:**

LoRA is the method of choice to finetune LLMs or any DN in general. This is typically done by SGD since the nonlinearity of the DN prevents any closed-form. The paper looks at an extremely simplified model--under which it is possible to obtain a closed-form solution. Having a closed-form solution is great, but as it can only be applicable to two-layer networks which are not used in practice, the actual impact of the study feels absent. It is also unclear if even within that space of models the method has any impact on challenging datasets.

** Strength **
- While I do not believe the considered setup can be useful in practice (see below), when looking only at this particular setup the work is novel
- The presentation provides clear mathematical results that can be used in practice--albeit in a non-realistic setup
- The need for more theory around LoRA is high hereby making the paper quite well timed with today's research problem

** Weakness **
- while there are some minor typos, and some grammatical issues, the entire writing is OK to follow
- the major issue is the limitation of the method to two-layer models which is extremely far from the LoRA setup with attention and multiple blocks. This is a key limitation since none of the results can be leveraged as of today in existing models--and insights from shallow networks do not extend to deep networks as well demonstrated by the numerous NTK and similar works.
- beyond the shallow network limitation, the datasets considered are extremely simple to solve for any algorithm and thus do not provide meaningful signal to assess the quality of the propose method
- lastly, the presentation is not friendly to practitioners from different fields which reduces the impact of the submission even further

**Strengths:**

Please see summary

**Weaknesses:**

Please see summary

**Questions:**

Please see summary

---

### Official Review · Reviewer_nCFn · 2025-11-01

**Soundness:** 3
**Presentation:** 3
**Contribution:** 3
**Rating:** 6
**Confidence:** 3

**Summary:**

The paper proposes a novel, non-iterative method for LoRA as an alternative to gradient-based fine-tuning. The method is restricted to two-layer neural networks and computes a closed-form solution in two steps:

Layer 1: It uses Stein's lemma to derive an analytical, closed-form estimator for the first-layer adaptation matrix $\hat{\Delta}_1$.

Layer 2: It solves for the second-layer adaptation $\hat{\Delta}_2$ using standard reduced-rank ridge regression.

The authors provide theoretical consistency guarantees under a projection adaptation assumption and claim the method is comparable to standard LoRA while being significantly faster.

**Strengths:**

1. The paper preposes Stein-based subspace recovery for layer 1 and reduced-rank regression for layer 2. This statistical analysis is novel and has a nice view. If the assumptions were to hold, the computational speedup is a significant practical benefit.

2. The paper demonstrates that its closed-form solution potentialy serves as a high-quality, non-zero initialization for standard gradient-based LoRA. This "SGD-I" baseline consistently outperforms standard LoRA with zero initialization .

**Weaknesses:**

1. The proposed method only applies to a simple two-layer feedforward network. This is not a general-purpose LoRA replacement and has almost no applicability to adapting modern LLMs. Framing as “consistent LoRA” feels incremental relative to existing dimension-reduction regression and reduced-rank literature.

2. The proposed method is limited to some restrictive assumptions. For example, it requires knowing the second-order score function $S(z)$ of the input representation $Z = \phi(X)$. This is impossible in a practical setting (e.g. for an LLM's hidden states). The paper circumvents this on MNIST only by assuming the latent space is Gaussian, which is a major simplification .

3. The paper lacks experiments that represent real world problems. The only "real-world" experiment is on MNIST which is a toy problem that does not support the paper's claims about LLM fine-tuning. Furthermore, the network being adapted is tiny (input dim 20, hidden dim 1024), which is not representative of any modern task.

**Questions:**

Please see Weaknesses section

---

### Official Review · Reviewer_9z6j · 2025-11-04

**Soundness:** 3
**Presentation:** 3
**Contribution:** 3
**Rating:** 4
**Confidence:** 3

**Summary:**

This work presents a non-iterative alternative to Low-Rank Adaptation (LoRA) for two-layer neural networks. By combining Stein’s lemma with reduced-rank ridge regression, the method analytically estimates low-rank adaptation parameters without iterative optimization. Theoretical consistency guarantees under a projection adaptation assumption are provided, and comparable accuracy to SGD-based LoRA is achieved while being orders of magnitude faster. The approach also serves as an effective initialization for iterative fine-tuning, offering a fast, theoretically grounded framework for efficient transfer learning

**Strengths:**

The theoretical guarantees are very complete, and the algorithm is very nice. The 2-layer structure is taken advantage of to the fullest extent. The experiments with 2-layer networks (and fixed feature maps) also consistently show improvement upon the standard SGD training.

**Weaknesses:**

I'm really unsure about the generalizability of the knowledge provided in this work. In practice, the neural networks will have more than 2 layers. The proposed training algorithm seems difficult to generalize to the setup of multiple layers (the authors mention that the generalization to 3 or more layers is possible when second-order score functions for hidden activations are available, but this seems like an unrealistic assumption).

I understand that deep learning theory often relies on stylized and restrictive assumptions, but the setup being studied should have some shared common structure with the practical setting. In this case, the setup being studied has a different architecture and a different algorithm compared to the standard deep network with LoRA modules trained via SGD.

**Questions:**

In what sense is the analysis of this work informative for the standard practical LoRA training?

---

### Note · Authors · 2025-11-19

I have read and agree with the venue's withdrawal policy on behalf of myself and my co-authors.